# An Explainable Markov Chain–Machine Learning Sequential-Aware Anomaly Detection Framework for Industrial IoT Systems Based on OPC UA

**DOI:** 10.3390/s25196122

**Published:** 2025-10-03

**Authors:** Youness Ghazi, Mohamed Tabaa, Mohamed Ennaji, Ghita Zaz

**Affiliations:** 1Pluridisciplinary Laboratory of Research and Innovation (LPRI), EMSI, Casablanca 20330, Morocco; y.ghazi@emsi.ma; 2Digital Engineering for Leading Technologies and Automation Laboratory (DELTA), ENSAM, Casablanca UH2C, Morocco; ennaji.moh@gmail.com (M.E.); zaz.ghita@ensam-casa.ma (G.Z.)

**Keywords:** ICS/SCADA security, OPC UA, anomaly detection, Markov chains, sequential modeling, machine learning, causal inference, explainable AI, IIoT edge

## Abstract

Stealth attacks targeting industrial control systems (ICS) exploit subtle sequences of malicious actions, making them difficult to detect with conventional methods. The OPC Unified Architecture (OPC UA) protocol—now widely adopted in SCADA/ICS environments—enhances OT–IT integration but simultaneously increases the exposure of critical infrastructures to sophisticated cyberattacks. Traditional detection approaches, which rely on instantaneous traffic features and static models, neglect the sequential dimension that is essential for uncovering such gradual intrusions. To address this limitation, we propose a hybrid sequential anomaly detection pipeline that combines Markov chain modeling to capture temporal dependencies with machine learning algorithms for anomaly detection. The pipeline is further augmented by explainability through SHapley Additive exPlanations (SHAP) and causal inference using the PC algorithm. Experimental evaluation on an OPC UA dataset simulating Man-In-The-Middle (MITM) and denial-of-service (DoS) attacks demonstrates that incorporating a second-order sequential memory significantly improves detection: F1-score increases by +2.27%, precision by +2.33%, and recall by +3.02%. SHAP analysis identifies the most influential features and transitions, while the causal graph highlights deviations from the system’s normal structure under attack, thereby providing interpretable insights into the root causes of anomalies.

## 1. Introduction

Modern ICS/SCADA networks must guarantee reliable, orderly, real-time communication between controllers, sensors, actuators, and supervision platforms. The OPC UA protocol is one of the essential communication solutions in industrial environments, facilitating remote monitoring and control thanks to its interoperability. However, its open design—one of the key enablers of Industry 4.0 [1]—also considerably enlarges the attack surface: Man-In-The-Middle (MITM) interceptions, command injections, replay attacks, and denial-of-service attacks are on the rise. While the ongoing shift towards Edge-IIoT architecture is complicating industrial cybersecurity. Each layer—from the sensor to the edge gateways, then to the aggregation networks and the cloud—introduces specific vulnerabilities such as weak authentication, incomplete encryption, unpatched legacy components [2,3,4,5]. Several recent incidents illustrate the severity of these risks, including the attack on critical Spanish infrastructure via a misconfigured OPC UA gateway [6]: the intrusion exploited the compromise of connection sequences and command transitions, even though the individual values of the variables were not out of the ordinary. In addition, recent analyses [7] reveal that more than 62% of exposed OPC UA servers suffer from configuration flaws or weak security policies, further aggravated by the heterogeneity of software implementations. In this context, the attack surface now extends from the field layer to application protocols, underscoring the need for multi-layered countermeasures. There is a need to develop sequential and explainable anomaly detection capable of reasoning about the state transition logic in OPC UA communications. Machine learning approaches are therefore promising, provided that (i) they explicitly integrate this sequential dimension to distinguish a contextual deviation from an isolated incident and (ii) they provide operational explainability to accelerate remediation in critical environments. From this perspective, our approach stands out from existing works, which typically rely either on purely statistical analysis without memory or sequential models considered in isolation. We propose a unified framework combining Markov memory, machine learning, and explainability. This hybridization makes it possible to capture the temporal dependencies specific to OPC UA communications, exploit the nonlinear relationships between variables, and provide a multi-level interpretation of the anomalies detected. It thus introduces a clear methodological innovation: the ability to reveal progressive or contextual attacks that escape conventional approaches, which are often limited to instantaneous signals or simple sequential patterns.

This raises two research questions:How can stateless machine learning models be effectively transformed to incorporate the sequential dynamics inherent in industrial protocols?How can anomalies be detected that reside not in static values, but in the structure and temporality of transitions?

To address these challenges, we present a hybrid pipeline that allows us to:(1)Model the OPC UA sequence using adaptive Markov chains.(2)Train several ML detectors (OCSVM, Isolation Forest, LOF, MLP) to capture various anomalies.(3)Integrate explainability and causal inference modules to attribute the impact of variables and reveal underlying dependencies.

We evaluate the proposed approach on a dataset representative of OPC UA traffic in an industrial context. The experimental results show a significant improvement in detection performance, particularly when sequential memory is injected into the models. After benchmarking and comparative testing of the different ML models studied, the Isolation Forest model achieves an F1-score of 0.986 without memory, rising to 0.993 with the addition of memory. Similarly, the MLP maintains a high F1-score ≈ 0.998, while gaining robustness and explainability thanks to sequential representation. In addition to the quantitative results, explanatory analysis by SHAP and causal inference reinforce the relevance of Markov hybridization. For the MLP and Isolation Forest models, the addition of the sequential dimension is not limited to an improvement in overall performance: it also induces a significant change in the internal decision-making mechanisms. From a causal perspective, structural dependency graphs show that “memory” variables play a causal intermediary role between several operational indicators and the “anomaly” output, illustrating a cause–effect path based on temporal dynamics. These results demonstrate that the integration of sequential memory is not only beneficial for performance, but also for the understanding and traceability of alerts, which is a critical requirement in sensitive industrial environments. The main contributions of this work are summarized as follows:Hybrid framework design: We propose a unified anomaly detection framework that combines adaptive Markov chains, multiple machine learning detectors, and explainability modules to explicitly capture temporal dependencies in OPC UA traffic.Sequential-aware enhancement: We demonstrate how stateless models such as MLP and Isolation Forest can be enriched with Markov-based memory to improve both accuracy and robustness in detecting contextual anomalies.Explainability and causality integration: We incorporate SHAP analysis and causal inference graphs to provide multi-level interpretability, enabling better traceability and faster remediation in industrial environments.Comprehensive evaluation: We validate our approach on a realistic OPC UA dataset with industrial attack scenarios (MITM, DoS), showing significant gains in F1-score and interpretability when sequential memory is injected.

The remainder of this article is organized as follows: Section 2 reviews related work. Section 3 formalizes the proposed model and describes the dataset. Section 4 details the experimental evaluation, including detection performance, SHAP-based explainability, and causal inference results. Section 5 discusses the main lessons learned from this study. Finally, Section 6 concludes and proposes perspectives for industrial deployments, particularly in edge environments.

## 2. Related Work

### 2.1. ICS Anomaly Detection Using Machine Learning

In recent years, numerous studies have explored intrusion detection systems (IDS) based on machine learning for industrial environments. Motivated by the ability of these models to identify unknown attacks without pre-established signatures, several approaches have been tested on SCADA/ICS test benches, including the SWaT [8], WADI [9], and BATADAL [10]. Algorithms such as One-Class SVM [11], Isolation Forest, LOF, and neural autoencoders [12] have demonstrated good detection performance. However, most detection systems currently in use are still based on stateless or quasi-stateless models, which evaluate each observation independently or via very short time windows, without exploiting the typical sequences of industrial protocols [13,14]. This limitation is more critical in IIoT architecture, where the sequential dimension and consistency of transitions are meaningful. As several studies [15,16] have pointed out, strictly static models struggle to detect advanced attacks that exploit temporal dependencies.

To address this shortcoming, some authors have proposed introducing memory mechanisms into detectors. For example, refs. [17,18,19] using Markov chains to model packet sequences in ICS flows, making it possible to identify subtle deviations from normal transitions. These approaches are effective in detecting stealthy attacks (slow scans, gradual variations), but they are also highly sensitive to benign anomalies related to background noise or natural variability. In addition, other recent work has explored combinations of sequential models and deep learning (e.g., LSTM or GRU networks) followed by downstream classifiers [20,21,22,23]. Although promising in terms of results, these solutions are often difficult to interpret and illustrate real-time and edge computing constraints [24,25,26].

IDS approaches based on machine learning continue to face important limitations in industrial contexts, particularly their inability to capture the sequential and behavioral dimensions of advanced attacks. Most of these systems analyze static features or isolated data snapshots, which makes them efficient but prevents them from linking events across time. As a result, they often miss malicious activities that evolve gradually or follow multi-step attack patterns [27,28]. This limitation is especially critical in IIoT settings, where intrusions rarely occur in a single action but unfold through carefully orchestrated sequences that traditional supervised algorithms fail to detect [29]. To address this gap, Markovian stochastic models have been introduced, explicitly representing the sequence of actions or system states. By estimating transition probabilities between states, Markov and Hidden Markov Models (HMM) integrate the temporal dimension of system behavior and support behavioral anomaly detection, which targets deviations from normal sequential patterns. This complements signature-based IDS, designed to match current activity with known attack profiles [30,31]. Combining both paradigms enables systems to detect novel behavioral patterns while continuing to recognize known attack signatures, thus improving robustness [32]. Nevertheless, basic Markov models remain constrained by their “short memory,” because they consider only a short span of past events. This drawback makes them prone to overlooking long-term dependencies, highlighting the need for hybridization with more advanced learning methods to detect sophisticated multi-stage attacks.

As highlighted in the literature, stateless models (e.g., Isolation Forest, MLP, Autoencoder, etc.) are valued for their ease of deployment, speed of execution, and, in some cases, explainability via post hoc techniques. Conversely, their ability to identify sophisticated attacks exploiting temporal dependencies remains limited. Conversely, explicit memory models (RNN, LSTM, GRU, Markov) better capture industrial dynamics, but face constraints in terms of complexity, interpretability, and computational cost on edge platforms. This overview emphasizes an unresolved challenge calling for the design of capable hybrid frameworks, integrating sequential memory while maintaining explainability and efficiency compatible with IIoT constraints.

From a technical standpoint, the different families of sequential models (RNN, LSTM, GRU) differ in terms of trade-offs between computational cost, memory footprint, and accuracy, which determines their deployment at the edge. A simple recurrent neural network is inexpensive to infer, requires few operations per time step, and is relatively memory-efficient, but it only stores short-term temporal dependencies. LSTM architecture introduces gate mechanisms that allow information to be retained or forgotten over long sequences, at the cost of greatly increased complexity: a typical LSTM has about four times more parameters than a standard RNN of equivalent size [33,34]. As a result, LSTMs consume more memory and induce higher inference latency due to additional calculations per cell than simple RNNs [35]. GRUs (Gated Recurrent Units) offer a compromise by simplifying the LSTM architecture—with only two gates instead of three—which reduces the number of parameters by about 25% while maintaining predictive performance close to that of LSTMs [36]. This reduction in complexity translates into a smaller memory footprint: an LSTM model occupied ~42 MB, compared to ~31 MB for its GRU equivalent, and faster training and inference times. GRUs typically train 20–30% faster than comparable LSTMs [37,38,39]. In terms of detection accuracy, LSTM/GRU models excel at capturing complex sequential patterns and thus achieve higher detection rates on long sequences than simpler approaches. In a comparative evaluation [40], a GRU achieved an F1-score of 0.91 compared to 0.92 for an LSTM, while requiring ~30% less computing time [41], illustrating that simplifying the architecture results in only a slight loss of accuracy for a substantial gain in efficiency. On the other hand, classic Markov models with limited state memory have virtually zero computing and storage requirements, but their predictive capacity is inferior for complex attack scenarios or long chains of actions, unless the order of the model is increased, which then exponentially increases the number of states to be considered.

By analyzing the limitations of existing approaches, three key weaknesses emerge:-Lack of temporal memory: context-limited models ignore long-term trends. This makes gradual attacks (e.g., slowly drifting sensors) or discrete message reordering invisible.-Vulnerability to fixed thresholds: ML models can be exploited by injecting low-level but structured noise, leading to misclassifications [42].-False positive/false negative trade-off: overly sensitive settings cause excessive alerts, while overly tolerant thresholds create a gray area that is conducive to stealth attacks [43].

Considering these findings, our work explores three main hypotheses:Does adding a Markovian memory score significantly improve anomaly detection by static models?Does a higher memory order enrich the quality of the information, or does it introduce harmful noise?Is there an identifiable causal link between memory order, Markov score, and the final decision of the ML model?

Our approach is consistent with this, combining the advantages of sequential modeling—to capture the temporal dynamics specific to industrial systems—and the analysis of quantitative variables derived from statistics, thus offering a compromise between detection performance, explainability, and efficient deployment on edge architectures. Considering this, the hybridization of Markov chains and machine learning is a methodological choice based on their complementarity. Markovian models effectively capture the local dynamics of transitions in OPC UA traffic by representing short memory and sequential dependencies, while ML models exploit statistical structure and nonlinear relationships between features. Taken in isolation, stateless ML models struggle to integrate temporal information, while purely sequential models remain limited in their ability to discriminate complex patterns in the variable space. Markov–ML hybridization therefore offers a robust compromise: it enriches anomaly detectors with explicit temporal context, while retaining the discriminative power and generalizability of learning algorithms, which is particularly suited to IIoT environments where anomalies often emerge from local combinations of sequential and statistical behaviors.

### 2.2. Explicability and Causality for ICS Cybersecurity

While most anomaly detection studies for ICS have focused on improving detection performance using traditional metrics (precision, recall, F1-score), interpretability is increasingly recognized as a key requirement for scalability and operational acceptance. When faced with an alarm, operators must be able to understand not only which variables triggered the detection, but also how temporal or sequential dynamics contributed to the decision, to assess the actual criticality of the event and take effective action [44]. Post hoc explainability methods such as SHAP have therefore been employed in machine learning models to quantify the contribution of each variable to anomaly detection in ICS. For instance, ref. [45] demonstrated that applying SHAP to an LSTM model enables the precise identification of sensors involved in attacks on physical processes, with the contribution of these variables explaining most alerts. However, these approaches remain essentially qualitative and are most often limited to the post hoc analysis of specific incidents.

Explainability can be leveraged not only to inform model decisions but also to guide the optimal choice of sequential memory length to be integrated into Markov–ML hybrid models. Some recent work [46] suggests that a joint analysis of SHAP scores and performance metrics can reveal the saturation point beyond which increasing the order of the Markov memory no longer provides any explanatory or operational gain. This paves the way for optimization strategies based on explainability that are both robust and interpretable, particularly suited to the constraints of edge platforms. In addition to explainable approaches, causal analysis provides an additional lever for the detailed interpretation of abnormal behavior, seeking to link detected anomalies to their potential causes through the structure of dependencies between variables. The PC (Peter and Clark) algorithm [47,48] is commonly used to infer directed causal graphs from data by identifying conditional independence relationships between variables. In the field of industrial systems, recent work [49] proposed an approach combining causal graphs and unsupervised detection on the SWaT and WADI datasets: they used the PC algorithm to model causal links between sensors under normal conditions, then identified anomalies as breaks in dependency in the graph, thereby locating the affected “causal slice” and the potential root cause.

As shown in the table below, stateless models have the advantage of being fast, easy to deploy, and often explainable via post hoc methods. However, their ability to detect sophisticated attacks by exploiting temporal dependencies or transition logic remains limited. Conversely, sequential models (LSTM, GRU, Transformers, Markov chains) can capture these dynamic patterns.

## 3. Methodologies

### 3.1. System Model

Figure 1 shows the overall architecture of the hybrid pipeline developed in this study. It is based on four complementary functional modules, designed to take advantage of both the sequential dynamics of OPC UA traffic and statistical analysis techniques:Markov module: captures the probabilistic transitions between OPC UA session states (e.g., StartRawConnection → SecureChannel → Session → ReadRequest, etc.) by constructing Markov chains of variable order. This module detects unusual deviations in the temporal structure of the flows.ML detectors: a set of machine learning models operating on classic statistical characteristics of traffic (bytes, duration, frequency, etc.), with and without sequential memory.SHAP explanation: for each alert triggered, this module identifies the characteristics that contributed most to the detection, via the assignment of SHAP values.Causal analysis: estimates a graph of causal dependencies between the observed variables, making it possible to explain the mechanisms underlying the detected anomalies (e.g., propagation effects, abnormal dependencies).

To combine the two sources of information (sequential and statistical), we define an overall anomaly score as the weighted average of the score produced by the Markov model (based on transition probabilities) and the score produced by the ML model. This strategy allows us to preserve the sensitivity of the Markov model to temporal irregularities, while benefiting from the robustness of ML models with respect to non-sequential variations.

### 3.2. Dataset OPC UA

The dataset used in this study is a real OPC UA traffic recording simulating a typical industrial network [50,51], containing both normal sessions and various forms of attacks, including Man-In-The-Middle attacks, packet injections, and controlled re-injections.

It includes 107,633 records and 32 explanatory variables, including network characteristics (IP addresses, ports), flow statistics (bytes, duration, frequency), and OPC UA protocol indicators (messages, sessions, errors). The target variable is binary: 0 for normal flows and 1 for abnormal flows.

As illustrated in Figure 2, the data is organized in the form of a structured three-dimensional matrix, according to three axes of analysis:

(i) the functional category of each variable (X axis),

(ii) its structural nature, either static or sequential (Y axis),

(iii) and its memory depth (Z axis).

Each bar shown in the figure corresponds to the occurrence of a variable at a given memory level.

This structure is based on a high degree of sequential consistency observed in service exchanges in OPC UA, particularly through sequences such as: Create Session → ActivateSession → Read Request, but also through numerical variables associated with durations, delays, and rates, which makes them particularly suitable for evaluating detection methods that integrate sequential memory with a longer length. Even though the OPC UA dataset used accurately reflects industrial traffic and provides a solid basis for experimental evaluation, certain limitations must be acknowledged. First, the attack scenarios included mainly cover classic threats such as “scans, denial of service, message injections, parameter alterations” and do not represent the full range of recent OT attacks, such as those implemented by new-generation malware such as Industroyer2 and PIPEDREAM [52,53]. Second, the distribution of classes may introduce bias: some attacks are overrepresented, while normal behaviors related to specific operational phases (startup, shutdown, maintenance) appear less frequently. Finally, the collection environment remains a controlled test bed, limiting the natural variability encountered in real-world conditions, where human error, network disruptions, and physical anomalies occur. These restrictions do not invalidate the usefulness of the dataset but must be considered when interpreting the results. Our models demonstrate robust generalization to statistical and sequential regularities, but their ultimate validity will need to be confirmed on more heterogeneous real-world flows and across a broader spectrum of attack scenarios.

### 3.3. Data Processing Procedure

To capture the dynamics of OPC UA exchanges, raw network data is transformed into discrete time sequences, suitable for probabilistic modeling using Markov chains. Formally, we represent each session as a sequence of discrete states:(1)$$S=\left(x_{1},x_{2},\ldots .,x_{T}\right)\;\;with\;x_{T}\in \;A$$
where each $x_{T}$ corresponds to a discrete value from a categorical variable of the network and A being the finite alphabet of states.

The division of sequences S is based on three main criteria, allowing the logical and temporal consistency of flows to be preserved:IP session: grouping of frames according to the source address (src_ip), simulating continuous interactions of the same automaton or sensor.TCP flow: segmentation based on session boundaries (flowStart, flowEnd) to reflect complete transactions.Time windowing: application of a sliding window to long sessions to capture recurring patterns localized in time.

Our goal is to provide the pipeline with explicit sequential memory, where the depth of the memory is controlled by order K:With order 1, each state is conditioned solely by the immediately preceding state.At order 4, the probability of the current state depends on the four previous states, thus incorporating a richer contextual history.

The Markov transition probability of order *K* is written as [54]:(2)$$P{(x}_{t}\left|\;x_{t-1},\;\ldots ,\;x_{t-k}\right)=\frac{C\;(x_{t-k},\;\ldots ,x_{t})}{C(\;x_{t-k}\;,\;\ldots ,x_{t-1})}$$
where C(x) denotes the number of occurrences of the sub-sequence x in S.

From this, we can deduce the associated transition matrix, denoted $T_{k}$, such that:(3)$$T_{k}\left[u,v\right]=P(x_{t}=v\;|x_{t-k}^{t-1}=u)$$
with $u\in A^{k}$ the current context, and v∈A the next state. This matrix constitutes a normal profile of observed sequences, against which any test sequence will be compared. A marked divergence between the observed probabilities and those of the model indicates a potential anomaly, even if each isolated event appears legitimate.

#### 3.3.1. Derivation of Markov-Based Characteristics

From each sequence or sub-sequence $\;S_{i}=(x_{1},\ldots ,x_{T})$, we extract several mathematical descriptors based on kth-order Markov modeling. These scores allow us to capture dynamic properties and identify structural breaks within the flow. 

Log-likelihood evaluates the probability of the entire sequence according to the learned model [55]:


(4)
$$L_{k}\left(S\right)=\sum_{t=k+1}^{T}\log P\;(x_{t}|x_{t-1,\ldots ,}x_{t-k})\;\;\;\;\;\;\;\;\;$$


Local conditional entropy measures the uncertainty about the next state given the current context:


(5)
$$H_{k}\left({\;x}_{t-k}^{t-1}\;\right)=-\sum_{x\;\epsilon \;A}P(x\left|x_{t-k}^{t-1}\;\right)\log P(x\;|x_{t-k}^{t-1})$$


Anomaly score (or “surprise”): quantifies the local improbability of a transition:


(6)
$${Surprise}_{k}\left(t\right)=-\log P(x_{t}|x_{t-1},\ldots ,\;x_{t-k})$$


A high surprise score indicates a rare transition, which may reflect abnormal behavior.

#### 3.3.2. Fusion of Classic and Sequential Features for Machine Learning

To exploit both classic statistical dimensions and sequential dynamics, we construct an enriched vector for each observation [56].(7)$$x_{i}^{\left(k\right)}=\left[\;z_{i\;};\;m_{i}^{\left(k\right)}\right]\;\epsilon \;R^{d+q}$$
where $z_{i}\epsilon R^{d}$ is the vector of classical features (throughput, duration, error rate, etc.), and $m_{i}^{\left(k\right)}\epsilon R^{q}$: is the vector of Markovian features of order k. The general state vector at time t aggregates several memory orders via:(8)$$X_{t}=[x_{t}^{\left(1\right)},\;x_{t\;\;}^{\left(2\right)},\;\ldots ,x_{t}^{\left(m\right)}{]}^{T}\;\in \;R^{m(d+q)}$$

We thus combine two families of complementary descriptors: instantaneous statistical features of the OPC UA traffic (throughput/load, timings, business ratios, errors/size), and sequential features derived from a Markov model of order *k*. Here, $z_{i}$ gathers the variables listed in Table 1, while $m_{i}^{\left(k\right)}$ encodes the historical context of depth *k*. This enriched vector then consistently feeds supervised, semi-supervised, or unsupervised models, enabling the simultaneous exploitation of both the protocol snapshot and the sequential dynamics of OPC UA traffic. From an empirical standpoint, the introduction of a short-term Markov memory results in a notable improvement in F1-score and stability of the AIC, reflecting a better trade-off between accuracy and parsimony.

#### 3.3.3. Class Imbalance

The OPC UA dataset has an atypical configuration (Figure 3): a majority of abnormal instances, which contrasts with traditional intrusion detection contexts. This situation complicates the training of semi-supervised models designed to learn exclusively from normal samples.

From a formal point of view, this dataset can be mathematically noted as follows: $D=\{(x_{i}$,$y_{i}){\}}_{i=1}^{N}$, with $x_{i}\in R^{d}$ representing the characteristics (statistical and sequential), and $y_{i}\in \{0,1\}$ l’étiquette (0 = normal, 1 = abnormal). The imbalance is quantified by the irregularity ratio:(9)$$\;\;\;\;\;IR=\frac{\left|\left\{i\;:\;y_{i}=0\right\}\right|}{\left|\left\{i\;:y_{i}=1\right\}\right|}$$

Several classic rebalancing approaches have been considered, such as:Oversampling (by duplication or via the SMOTE algorithm [57]);Undersampling of the majority class [58];Class weighting in the loss function [59].

However, in our context, certain methods may introduce bias. Artificial oversampling is inappropriate for models trained exclusively on normal data, as it risks generating synthetic behaviors that are not representative of real system dynamics. We therefore adopt a hybrid strategy:Stratified subsampling of abnormal instances, to maintain a representative sample of the normal class.Weighting of the loss function for supervised models (MLP, LOF), expressed as [60]:(10)$$\;L_{weighted}=-\sum_{i=1}^{N}w_{yi}.\log P\left(y_{i}\right|x_{i})$$
with $w_{0}=\frac{1}{\left|\left\{i:y_{i}=0\right\}\right|}$, $w_{1}=\frac{1}{\left|\left\{i:y_{i}=1\right\}\right|}$. Finally, Markov scores of orders 1 to 4 are integrated as additional input variables to reinforce the temporal component of the sequences and improve the models’ ability to identify abnormal behavior in dynamic contexts.

### 3.4. Data Preprocessing

Data preprocessing is a crucial step in ensuring the quality, consistency, and comparability of the inputs used in anomaly detection. It involves systematically extracting and unifying two types of descriptors:Static characteristics (instantaneous descriptors, global statistics).Sequential characteristics (Markov transition scores, surprise indices, conditional entropy), capturing the temporal and contextual dynamics of the monitored process.

In addition, managing class imbalance—already discussed in Section 3.3—is essential at this stage to avoid bias during machine learning. The objective of this preprocessing is therefore to ensure alignment between static and sequential vectors, transforming them into a unified and homogeneous input space for all models in the pipeline.

#### 3.4.1. Standardization of Numerical Variables

To ensure scale homogeneity and improve model convergence, all numerical variables $x_{t}^{\left(j\right)}$ are standardized [61]:(11)$$X_{t}^{\left(j\right)}=\;\frac{x_{t}^{\left(j\right)}-\mu_{j}}{\sigma_{j}}$$
where $\mu_{j}$ is the empirical mean of variable j, and $\sigma_{j}$ is the associated standard deviation.

This transformation ensures that:(12)$$Ε\left[X_{t}^{\left(j\right)}\right]=0,Var\left(X_{t}^{\left(j\right)}\right)=1$$

Thus, allowing for a fair comparison between heterogeneous characteristics.

#### 3.4.2. Encoding Categorical Variables (One-Hot Encoding)

Non-numerical variables (e.g., proto, service) are transformed by one-hot encoding, according to:(13)$$cat\;\left(x_{t}\right)\;\epsilon \;\left\{c_{1},c_{2},\ldots ,c_{C}\right\}\;\Rightarrow OH\;\left(x_{t}\right)=[\;0,\ldots .,1,\;\ldots ,0{]}^{T}\;\epsilon \;\{0,1{\}}^{C}$$

The final vector $X_{t}\;\epsilon \;R^{m^{\prime }}(with\;m^{\prime }=m+C)$ is then obtained by concatenating the standardized numerical variables and the encoded categories:(14)$$X_{t}=\left[\;\underset{Standardization\;}{\underbrace{\frac{x_{t}^{\left(1\right)}-\;\mu_{1}}{\sigma_{1}},\ldots ,\;\frac{x_{t}^{\left(m\right)}-\;\mu_{m}}{\sigma_{m}};\;\;\;}}\;\;\underset{Categorical\;encoding\;}{\underbrace{OH\;\left(\;x_{t}^{\left(proto\right)}\right),\;OH\;(x_{t}^{\left(service\right)})}}\;\right]$$

This vector summarizes the overall state of the system at a given moment and constitutes the input for anomaly detection models.

#### 3.4.3. Integration and Selection of Characteristics

After generating statistical and Markovian descriptors (orders m = 1…4), a merged data matrix is constructed:$$\;X=\left[Z,\;\;Statistics,\;MarkovScore1,\;MarkovScore2,\ldots ,MarkovScorek\right]\in R^{n\times d}$$
where Z represents the classical variables, n is the number of samples, and d is the total number of features.

The following steps are applied:

Variance filtering: application of a threshold ε via VarianceThreshold to eliminate columns with too low variance; retain only features j such that $Var(X^{\left(j\right)}<\varepsilon )$.Imputation of missing values: each missing value $x^{\left(j\right)}$ is replaced by the median of the corresponding column:(15)$${\;\;x}^{\left(j\right)}\leftarrow Median\left(X^{\left(j\right)}\right)\;$$Normalization: to limit the influence of extreme values, normalization based on the median and interquartile range (IQR) is applied:


(16)
$$X^{\left(j\right)}\leftarrow \frac{X^{\left(j\right)}-Median(X^{\left(j\right)})}{IQR(X^{\left(j\right)})}\;\;\;\;$$


Once preprocessed and normalized, the data are fed into the different models of the pipeline, enhancing their robustness against noise, the inherent variability of industrial processes, and the imbalance in class distribution.

### 3.5. Detection Models and Theoretical Foundations

#### 3.5.1. Learning Paradigms

Our benchmark is based on the comparative evaluation of models representative of the main learning paradigms used in industrial anomaly detection:Unsupervised learning:

The models are assigned an anomaly score to each observation without using labels, based on density or isolation in the data space:$$Μ:R^{q+d}\to \left\{score\right\}\;(e.g.,Isolation\;Forest)$$

Semi-supervised learning:

The algorithm is trained only on normal data (majority class) and detects any behavioral deviation:$$f:R^{q+d}\to \left\{-1,+1\right\}\;(e.g.One-Class\;SVM)$$

Supervised learning:

Models learn a classification function from labeled data:$$f:R^{q+d}\to \left\{0,1\right\}\;(e.g.Random\;Forest,\;MLP,\;KNN)$$

We prioritize Isolation Forest, Random Forest, and Multi-Layer Perceptron because this trio satisfies three structural constraints of the Edge-IIoT context: (i) very low latency and memory footprint during inference [62,63], (ii) native compatibility with the short sequential memory injected by our Markov–ML pipeline, and (iii) coverage of heterogeneous label regimes: RF/MLP for supervised learning when reliable labels exist, IF for unsupervised learning when anomalies are rare or poorly labelled [64,65]. Specifically, the pipeline orchestrates a Markov module, which captures transition dynamics, and a family of ML detectors operating on statistical features enriched with short memory, thus combining temporal context and OPC UA protocol snapshots. Among the complementary models, OCSVM and LOF are retained as representative baselines for the semi/unsupervised families, but the focus is on IF/RF/MLP, whose precision/complexity ratio and material frugality are better suited to on-device deployment. It should be noted that Isolation Forest stands out for its ability to quickly isolate atypical observations in large industrial sets, while RF and MLP effectively model nonlinearities. Finally, while deep sequential architectures such as LSTM, GRU, and Transformers remain powerful for real-time modeling [66], their computational cost is prohibitive at the edge; our strategy is therefore to give lightweight models a short memory to capture the essence of the dynamics without degrading responsiveness at the edge of the network.

#### 3.5.2. Mathematical Foundations and Detailed Models

##### Isolation Forest

Isolation Forest is a random partitioning algorithm designed to isolate rare observations with fewer divisions [67,68]. Its principle is that anomalies are more easily separable than normal data [69].

The anomaly score $s\left(x\right)\;$ is defined by:(17)$$s\left(x\right)=2^{-\frac{E(h\left(x\right))}{c(n)}}$$
where $E(h\left(x\right))$ is the average separation depth of x, and c(n) is a normalization factor given by:(18)$$c\left(n\right)=2H\left(n-1\right)-\frac{2\left(n-1\right)}{n}$$
with $H\left(i\right)\approx ln(i)+\gamma$ (γ ≈ 0.5772 being Euler–Mascheroni’s constant). The closer the score $s\left(x\right)$ is to 1, the more abnormal the instance is. This model is robust, fast, and particularly well suited to large industrial datasets.

##### One-Class SVM

The One-Class Support Vector Machine (OCSVM) is a semi-supervised method designed to delineate the envelope of normal data by learning a boundary in a high-dimensional space to isolate atypical observations [70,71].(19)$${min}_{\;w,\;\;\rho ,\;\xi_{i}}\;\frac{1}{2}\;|\left|w\right|{|}^{2}+\frac{1}{\nu n}\;\sum_{i=1}^{n}\xi_{i}-\rho$$
Under constraints:(20)$$\left(w\cdot \phi \left(x_{i}\right)\right)\geq \rho -\xi i,\xi i\geq 0$$
where ϕ(⋅) is a nonlinear projection function in a reproducing Hilbert space, and ν∈[0,1] is a regularization parameter, controlling the maximum proportion of allowed anomalies and the minimum number of support vectors.

The final decision function of the model is given by:(21)$$f\left(x\right)=sgn\left(\sum_{i=1}^{n}\alpha_{i}K\left(x_{i},\;x\right)-\rho \right)\;$$
with $K\left(x_{i},\;x\right)$ a kernel function, most often Radial Basis Function (RBF). This approach has several advantages in an industrial context:It is trained solely on normal data, making it particularly suitable for contexts where anomalies are very rare.It is robust to noisy data if the parameter ν is well calibrated.It enables accurate detection of structural deviations even in large spaces.

##### Elliptic Envelope (Robust Covariance Estimation)

The Elliptic Envelope model assumes that normal data follows a multivariate Gaussian distribution. It relies on robust estimation of statistical parameters to reduce the influence of outliers. This method is well suited to the detection of multivariate anomalies, particularly in noisy industrial environments [72].

For each observation x, the robust Mahalanobis distance is calculated as follows:(22)$$DM(x)=\sqrt{(x-\;\mu {)}^{T\;}\;\sum_{.}^{-1}\left(x-\;\mu \right)}$$
where μ and Σ are the robust mean and covariance matrix. A decision rule is then applied using a threshold based on a quantile q of the empirical distribution of DM(x), to classify observations as normal or abnormal:$$x\;\in anomalie\;\Leftrightarrow D_{M}\left(x\right)>q$$

This model is particularly relevant for detecting consistent multivariate deviations, for example, during slow drifts on several sensors simultaneously, which may be overlooked by univariate or unstructured models.

##### Multi-Layer Perceptron (MLP)

The MLP (Multi-Layer Perceptron) is a deep neural network capable of learning complex nonlinear relationships between variables. It consists of fully connected layers, in which each neuron in one layer is connected to all neurons in the next layer. This model is particularly well suited to binary classification in industrial anomaly detection systems. Forward propagation through the layers is described by [73]:(23)$$a^{\left(l+1\right)}=\sigma (W^{\left(l\right)}a^{\left(l\right)}b^{\left(l\right)})$$
where $a^{\left(0\right)}=x\in R^{d}$ is the input vector (statistical and Markovian features), σ(.) is a nonlinear activation function (e.g., ReLU, tanh), and $W^{\left(l\right)}\;\in \;R^{n_{l+1}\times n_{l}}\;$ And $b^{\left(l\right)}\in \;R^{n_{l+1}}$ are the weights and biases of layer l. Learning is performed by gradient descent on a weighted cross-loss function (see section on imbalance), with parameters updated using the backpropagation algorithm.

##### K-Nearest Neighbors (KNN)

The KNN model is a distance-based supervised learning algorithm commonly used as a baseline in classification tasks. It assumes that observations that are close in feature space share similar labels [74,75].

For a new observation $x\in \;R^{d}$, the algorithm:Calculates the distance between x and all instances in the training set (often the Euclidean distance),selects the K closest neighbors $N_{k}\left(x\right)$,assigns the majority class among these neighbors.

The prediction is therefore given by:(24)$$\;y=mode\;\left(\left\{y_{i}|x_{i}\;\epsilon \;N_{k}\left(x\right)\right\}\right)\;$$
where $N_{k}$ represents the set of the K closest neighbors to x, and $y_{i}\;\in \{0,1\}\;$ are their respective labels.

The KNN is included in this study as a supervised base model in order to quantify the impact of integrating sequential features into an algorithm that, by nature, does not exploit them. This highlights the value of memory-based methods for anomaly detection in industrial systems.

##### Random Forest

Random Forest is an ensemble algorithm based on the construction of many independent decision trees, trained on random subsamples of the dataset and a random selection of variables at each internal split. It combines the predictions of these trees to improve robustness, generalization, and resilience to overfitting [76].

Given a set of T  trees $\{h_{t}(x){\}}_{t=1}^{T}$ each trained on a bootstrap of the data and a random selection of features at each split, the prediction for an observation $x\in R^{d}$ is given by:(25)$$\;\;\hat{y}\left(x\right)=majorite\prime \{h_{t}\left(x\right){\}}_{t=1}^{T}$$
where each tree $h_{t}\;$ votes for a class, and the final prediction is obtained by majority vote or average. The diversity introduced by bootstrapping and the selection of subsets of variables at each node reduces the correlation between trees and improves the overall stability of the model.

#### 3.5.3. Decision and Evaluation

##### Binarization of the Anomaly Score

After processing each model, a continuous score function $s_{M}\left(x\right)\epsilon R$ is produced for each observation x. In order to make this output usable in a real−time industrial context, a binary decision function is defined using a threshold τ, according to(26)$$\hat{y}\left(x\right)=I\left\{s_{M}\left(x\right)\;\geq \tau \right\}$$

The threshold τ is determined according to two main strategies:Quantile method: selection of a threshold based on the empirical distribution of scores (e.g., 95th percentile).ROC curve optimization: selection of the threshold that maximizes the trade-off between TPR and FPR (e.g., Youden’s criterion).

This transformation allows for an explicit separation between observations considered normal ($\;\hat{y}\left(x\right)=0)$ and those considered abnormal $\left(\hat{y}\left(x\right)=1\right)$.

##### Evaluation Metrics

The evaluation is based on classic binary classification metrics, defined from the elements of the confusion matrix [77]:-TP (True positives): anomalies correctly detected,-FP (False positives): normal cases incorrectly reported as anomalies,-TN (True negatives): normal cases correctly detected,-FN (False negatives): undetected anomalies.

The following metrics are calculated from these quantities:(27)$$(a)\;\mathrm{Precision}:\;\;\;Precision=\;\frac{TP}{TP+FP}$$

Proportion of alerts that are abnormal. A good indicator of alarm reliability.(28)$$(b)\;\mathrm{Recall}:\;\;\;Recall=\frac{TP}{TP+FN}$$

Detection rate of actual anomalies. Crucial in security-sensitive contexts.(29)$$(b)\;\;\;\;F1\text{-}\mathrm{Score}:\;F1=\frac{2.Precision.Recall}{Precision+Recall}\;$$(30)$$(c)\;F_{\beta }\text{-}\mathrm{Score}\;(\mathrm{recall}\;\mathrm{weighting}):\;F_{\beta }=\left(1+\beta^{2}\right).\frac{Precision\;.Recall}{\left(\beta^{2}.\;\;Precision\;\right)+Recall}$$

It gives more importance to recall if β >1(31)$$(d)\;\mathrm{Accuracy}:\;\;\;\;Accuracy=\frac{TP+TN}{TP+TN+FP+FN}$$

Proportion of correct predictions. Less relevant in unbalanced contexts.

(e) AUC-ROC and AUC-PR:
-AUC-ROC: area under the ROC curve, which plots TPR vs. FPR for all thresholds τ-AUC-PR: area under the Precision-Recall curve, more suitable for unbalanced datasets.

The metrics used—accuracy, recall, and F1-score—meet specific constraints for industrial anomaly detection. Accuracy measures the reliability of alarms by limiting false positives, which is critical in a monitoring environment where an excess of alerts can overwhelm operators and reduce confidence in the system. Recall quantifies the detector’s ability to avoid missing real anomalies, which is essential in industrial cybersecurity, since each false negative can correspond to an undetected intrusion with serious operational consequences. The F1-score, as the harmonic mean of precision and recall, captures the balance between these two requirements and provides a robust indicator when classes are unbalanced—a typical case in industrial scenarios where abnormal events are rare compared to normal traffic [78,79].

### 3.6. Parsimony and Complexity: Akaike Information Criterion (AIC)

To evaluate a model’s ability to generalize while remaining lightweight, the Akaike Information Criterion (AIC) is used [80]:

(a) Standard formulation:(32)$$AIC=2k-2ln(\hat{L})$$
where k the number of free parameters of the model, and $\hat{L}$ the maximum value of the likelihood function obtained after fitting the model to the data.

(b) Approximation using residuals:(33)$$AIC=2k+n\ln\left(\frac{RSS}{n}\right)$$
with $RSS=\sum_{i=1}^{n}{\left(y_{i}-{\hat{y}}_{i}\right)}^{2}$, where $y_{i}$ is the true class and ${\hat{y}}_{i}$ the model’s prediction.

The integration of the AIC score provides further insight into the suitability of models for deployment in edge environments, where lightness and computational efficiency are essential.

### 3.7. Training Protocols and Overfitting Control

#### Overfitting Control and Training/Validation Protocols:

In accordance with the recommendations in the literature, we have incorporated both specific safeguards for each model family and a transparent experimental protocol to ensure robustness and reproducibility. The data is divided into three distinct sets: 70% training, 15% validation, and 15% independent testing. This distribution is systematically respected and coupled with cross-validation [81,82] to stabilize the hyperparameters. The strategies applied are as follows:Isolation Forest—random subsampling, enough trees, calibration of the contamination rate by cross-validation, and selection of the model that minimizes inter-fold variance to limit overfitting and stabilize isolation scores.Random Forest—systematic use of out-of-bag error as an unbiased estimator of generalization, control of tree capacity, and maximum depth, adjusted by cross-validation [83].MLP—L2 regularization, dropout, and early stopping on validation, supplemented by Bayesian search for layer size hyperparameters, dropout rate, regularization coefficient, recognized practices to limit co-adaptation of neurons and stop training before validation loss drift.OCSVM—prior standardization and grid search of parameters ν, γ with RBF kernel, based on the margin/outlier rate compromise validated on ROC-AUC, which avoids an overly specialized boundary.LOF and k-NN—robust selection of k by cross-validation, distance weighting, and multi-k aggregation to reduce sensitivity to a single configuration, with the classic Cover–Hart results [84] motivating an increase in k to mitigate overfitting.Elliptic Envelope—robust estimation of mean/covariance via Minimum Covariance Determinant with choice of fraction h, which limits the influence of extreme points and local overfitting.

Finally, at the Markovian meta-model level, the memory order K is bounded by the AIC to avoid over-parameterization, and the decision thresholds are set on validation (Youden index on ROC) rather than on the training set. The reported results correspond to the mean and standard deviation calculated over ten independent repetitions, thus reinforcing the statistical robustness and reproducibility of the experimental setup.

### 3.8. Explicability

To ensure interpretability of the decisions, we integrated the SHAP method at every stage of the evaluation. This choice is grounded in solid theoretical foundations. Unlike other explainability techniques—such as Partial Dependence Plots, LIME, or Integrated Gradients—SHAP relies on cooperative game theory and provides a unique and consistent measure of importance for each variable, regardless of the interactions or the type of model employed, static or sequential. Shapley values guarantee three essential properties for faithfully interpreting algorithmic decisions: uniqueness, symmetry, and additivity. Several recent studies have demonstrated the superiority of SHAP in terms of stability and robustness of explanations in industrial and critical contexts [85]. This scientific foundation makes SHAP a preferred choice for ensuring transparency and trust in anomaly diagnosis.

For any observation $x_{i}$ characterized by a feature vector $X_{i}^{\left(K\right)}=[z_{i}\;,m_{i}^{\left(k\right)}]$ combining classical statistical features and sequential Markov scores of order $z_{i}$ and sequential Markov scores of order K, $m_{i}^{\left(k\right)}$, the model prediction f is accompanied by an additive decomposition of the feature contributions:(34)$$f\left(x_{i}\right)=\phi_{0}+\sum_{j=1}^{d+q}\phi_{i}\left(x_{i}\right)$$
where $\phi_{0}$ is the expected value of the model, and $\phi_{i}$ is the contribution of the j-ième feature to the individual prediction $f(x_{i})$. For each input variable (including Markov scores from order 1 to 4), the SHAP value is calculated according to the Shapley value formula:(35)$$\phi_{j}\;\left(x_{i}\;\right)=\sum_{S\subseteq F\;\setminus \left\{j\right\}}\frac{\left|S\right|!\left(\left|F\right|-\left|S\right|-1\right)!}{\left|F\right|!}[\;f_{\;S⋃\left\{j\right\}}\left(x_{i}\right)-f_{\;S}\left(x_{i}\right)\;]$$
where F is the set of features, and $f_{S}\left(x_{i}\right)$ the model prediction restricted to the subset S.

In our pipeline, this approach allows:Identification, for each anomaly score, of the explanatory share of sequential Markov features compared to standard variables.Comparison of the relative importance of different Markov memory orders within the decision process.Visualization of the actual influence of memory on the ML model’s decision.

### 3.9. Causal Structure Discovery

The objective is to automatically uncover cause-and-effect relationships between observation variables (traffic features, errors, time, etc.) and the occurrence of anomalies using the PC Algorithm [86]. The PC Algorithm (Peter–Clark) is a conditional independence algorithm that estimates the structure of a directed acyclic graph (DAG).

Inputs: normalized data matrix, with the option to include anomaly labels as the target variable.

$$\;D=\{X_{1}\;X_{2},\;\ldots .\;X_{P},Y\}$$
where Y is the target variable, and $X_{i}$ the features.

The output of the PC algorithm is a DAG G=(V,E) where V represents the variables and E the causal links. We model the dependence between variables using a directed causal graph:G=(V,E)
where $V=\{X_{1},X_{2}\ldots ,X_{d}\}$ is the set of system variables/features (network statistics, Markovian scores, etc.), and E⊂V×V is the set of directed edges representing direct causal relationships between variables.

The PC Algorithm uses conditional independence tests to build G.

For each pair of variables $\left(X_{i},X_{j}\right)$, conditional tests of the following type are performed:$$X_{i}⊥X_{j}|S⟹There\;is\;no\;direct\;edge\;between\;X_{i}\;et\;X_{j}$$
where:

$S\subseteq V\setminus \{X_{i},X_{j}\}$ is a conditioning subset,⊥ denotes conditional independence.

Thus, a causal link is absent if there exists a subset S such that $X_{i}$ and $X_{j}$ are conditionally independent. This allows for the removal of non-causal links using the separation properties in the graph.

Effect of an intervention (noted do-calculus):(36)$$Effet\left(X_{i}\to Y\right)=E\;[Y|do(X_{i}=x)]-E[Y|do(Xi=x^{\prime })]\;$$
where $do\left(X_{i}=x\right)$ denotes an intervention on $X_{i}$ (not just a simple observation), and E[] denotes the expectation under the intervened distribution.

This effect quantifies the expected change in the target Y resulting from a forced manipulation of variable $X_{i}$.

To ensure multi-level interpretability, we have integrated SHAP and causal inference. SHAP provides an additive decomposition of predictions into local and global contributions of input variables, regardless of the model used. For tree-based models (Isolation Forest, Random Forest), it highlights the characteristics that determine isolation scores or partition construction. In the case of neural networks, it weighs the hidden layers to quantify the contribution of the variables. For margin- or distance-based models, it directly translates proximities or isolations into local explanations, while for elliptical estimators, it identifies the variables that determine membership in a robust Gaussian distribution. In addition, the causal module uses the PC algorithm to infer the structure of dependencies between variables and highlight plausible directional relationships.

## 4. Results

This section presents the results of the experimental evaluation of the anomaly detection models, based on F1-score, precision, recall, AUC-ROC metrics, as well as the Akaike Information Criterion (AIC), which is used to estimate the trade-off between predictive performance and structural complexity.

The main objective is to analyze how the introduction of Markovian memory into stateless models influences their effectiveness. Specifically, we investigate from which order this sequential memory improves performance and from which point it instead introduces noise.

### 4.1. Overall Results and Performance Evolution

The comparative performance evaluation is conducted by training various detection models, both stateless and Markov-hybrid, using a combined feature set that includes statistical attributes as well as sequential representations extracted from industrial network data. Figure 4 illustrates the overall experimental pipeline, from data ingestion to result interpretation. Two experimental branches are presented: one dedicated to traditional memoryless models (Random Forest, Isolation Forest, MLP, etc.) and the other to hybrid versions that integrate Markovian memory with orders ranging from 1 to 4. This diagram structures the entire analysis protocol, highlighting the central role assigned to the impact of temporal dynamics in anomaly detection. After extracting statistical and sequential feature vectors from the dataset, these representations are merged to train each selected model. This step allows for comparison of their respective performances, while also providing a differentiated analysis of their interpretability capabilities via SHAP and their causal structure via the PC algorithm.

For each model, a binarization of the anomaly scores was applied (see Section 3.6) to calculate the various standardized metrics.

The confusion matrices (Figure 5 and Figure 6) provide a detailed view of the results for each model/order combination, highlighting variations in true positive and false positive rates.

It can be observed that:Random Forest and IF maintained excellent stability across all orders (low FP rate),OCSVM showed a significant decline in precision as early as order 1,MLP and KNN gained slightly in recall up to order 2, before declining thereafter.Figure 6 summarizes the overall results by showing the evolution of the F1-score according to the memory orders (from 0 to 4). It clearly shows:A moderate gain for MLP, KNN, and IF up to order 2 (+1–2.3%),stabilization or even a slight loss beyond that, indicating a memory saturation effect.A clear degradation for OCSVM, illustrating its poor adaptability to sequential structures.

The analysis of these results (see Figure 7) reveals behavioral heterogeneity among the models. Some models, such as Random Forest, are barely affected by the addition of memory, since their ensemble structure already captures complex patterns. Others, such as KNN or MLP, benefit from shallow memory to capture temporal regularities. Finally, hypersurface models like OCSVM are disrupted by nonlinear sequential dynamics, suggesting a structural incompatibility between their optimization principle and the sequential nature of enriched data. These results motivate a deeper exploration by model family to identify the factors explaining their sensitivity to the injection of sequential memory.

Decision Tree-Based Models:Random Forest (RF):

The performance of RF is generally stable and very high across the entire range of Markov orders (F1 ≈ 0.964–0.967). Order 2 yields the highest F1, with a precision of 0.973 and a specificity of 0.9878. This suggests that this model, already structurally robust, benefits marginally from short Markov memory order 1–2 but also tolerates the absence of memory just as well (Table 2).

Summary: Slight improvement of +0.4% in F1-score, followed by stabilization, and a minor decrease (<0.3%) beyond that.

Partitioning/Density Models

Isolation Forest (IF):

The IF model shows a slight but consistent improvement in F1-score between order 0 (F1 = 0.912) and orders 1–2 (F1 ≈ 0.914), before slightly declining at higher orders (F1 = 0.905 at order 4). This trend suggests that integrating short-term memory (order 1 or 2) is beneficial for detecting weak sequential patterns, but increasing memory complexity beyond that point no longer brings significant gains (Table 3).

Summary: Optimal gain of +0.25% in F1-score at orders 1–2, then a decline of ~1% due to overfitting (order 4).K-Nearest Neighbors (KNN):

KNN, on the other hand, significantly benefits from the addition of Markov memory up to order 2. The F1-score increases from 0.879 (order 0) to a maximum of 0.895 (order 2), a relative gain of +1.8%, before dropping back to 0.880 at order 4. This indicates that the algorithm makes good use of local sequential regularities, but beyond a certain threshold (order > 2), the neighborhood distance becomes less reliable due to increasing noise. Thus, the KNN model appears particularly responsive to moderate Markov memory, but sensitive to sequential overfitting (Table 4).

Summary: Moderate gain of +1.8% (order 2), then a loss of 1.6% (order 4).

Neural Models/Stateless ML Models

Multi-Layer Perceptron (MLP):

The MLP model shows a progressive improvement in F1-score up to Markov order 2 (F1 = 0.9413), before slightly decreasing at higher orders (F1 = 0.9334 at order 4). This indicates that injecting moderate sequential memory (orders 1–2) enhances the network’s ability to capture temporal dynamics, especially for diffuse or repeated attack patterns. The recall increases from 0.9086 to 0.9206 at order 2, illustrating improved anomaly coverage, while precision slightly decreases from order 2 onwards, betraying a gradual increase in false positives. Specificity falls slowly but steadily, indicating increasing sensitivity to variations deemed atypical by the model (Table 5).

Summary: Moderate gain of +0.31% (order 1), then stabilization at 0% (order 4).

Statistical Models (Adaptive Threshold Methods):

One-Class SVM (OCSVM):

The OCSVM model shows a clear and gradual decline in performance as the Markov order increases. The F1-score drops from 0.876 at order 0 to 0.799 at order 4, representing a relative decrease of 8.8%, accompanied by a drop in precision (from 0.961 to 0.906). This deterioration indicates that the model, which is naturally sensitive to noise and not robust to redundancies, is negatively affected by the addition of memory. The AIC also worsens, reaching 200.1 at order 4. We can deduce that OCSVM is not well suited for direct hybridization with Markovian memory in this context. There is a strong degradation of Recall (and thus F1) each time a Markov dimension is added, despite good Precision. At order 4, we obtained: Precision = 0.957, Recall = 0.612, F1 = 0.746, and AUC_ROC = 0.622. The AIC increases (from 14.6 to 20.6), indicating higher complexity and less model parsimony (Table 6).

Summary: Continuous degradation of F1-score: −5 to −9% as soon as Markov memory is added. No beneficial effectRobust Covariance (RobustCov):

The RobustCov model remains insensitive to the increase in Markovian memory, with stagnant performance (F1 ≈ 0.704, recall ≈ 0.587). No gain is observed beyond order 0, and the AIC remains unchanged. This stability suggests that this model, designed to detect extreme points in a global distribution, does not effectively exploit sequential information. This confirms its inadequacy in the case of contextual attacks where the order of events is critical (Table 7).

Summary: No gain, loss of ~1.3% from order 1. Curves remain completely stable thereafter.LSTM and GRU

With a view to methodological consolidation, the evaluation of MLP coupled with a second-order Markov memory was supplemented by a comparison with more expressive recurrent networks, in particular LSTM and GRU (Table 8).

This approach aims to evaluate the ability of these architectures to explicitly model the temporal dynamics underlying OPC UA flows, where sequence memory plays a central role in anomaly detection. We trained two compact variants of LSTM and GRU, each consisting of two stacked layers with a hidden dimension of 64 neurons and a dropout regularization mechanism set to 0.2. These models are followed by a binary linear classification head, activated by a sigmoid function to produce probabilities of belonging to the normal or abnormal class. Optimization is based on a weighted Binary Cross-Entropy (BCE) cost function, in which positive weights are estimated directly on the training set. This choice compensates for the marked imbalance between normal classes and anomalies, a phenomenon typical of industrial datasets.

The results shown in Table 8 indicate that the MLP approach enriched by a second-order Markov memory already achieves a high level of performance F1 = 0.966, confirming the value of explicit sequential modeling of local dependencies. However, the LSTM and GRU recurrent architectures slightly exceed this baseline, with F1 scores of 0.972 and 0.970, respectively. This relative superiority can be explained by their ability to learn directly from hierarchical and dynamic representations of OPC UA flows, capturing not only local transitions but also longer-range temporal dependencies. However, this gain in accuracy is accompanied by a substantial increase in memory usage—approximately 880 MB for LSTM and 860 MB for GRU—illustrating the well-known trade-off between performance and computational resources. In practical terms, this means that in industrial environments where resources are limited, the Markov-enhanced MLP provides a reliable and efficient choice. Conversely, when more powerful hardware is available, recurrent networks such as LSTM and GRU may offer a modest performance advantage worth exploiting.

### 4.2. Comparative Analysis of the Akaike Information Criterion

The analysis of the Akaike Information Criterion (AIC) as a function of Markov order reveals contrasting dynamics across model families (see Figure 8). This criterion, which penalizes excessive complexity while rewarding predictive effectiveness, provides a relevant indicator for assessing the trade-off between accuracy and parsimony.

Multi-Layer Perceptron: The AIC score decreases significantly between order 0 (5088.2) and order 2 (4715.4), reflecting a clear improvement in the quality/complexity trade-off thanks to the addition of sequential memory. From order 3 onwards, the curve flattens (4743.8), indicating a plateau effect.Random Forest: The AIC follows a similar trend, reaching a minimum at order 2 (4138.5) from an initial score of 4262.4. This suggests that the model effectively benefits from short-term memory, without additional overhead beyond that point.Isolation Forest (IF): There is a steady decrease in AIC up to order 2 (4757.8), but improvements become negligible from order 3 onwards, showing moderate usefulness of temporal memory.K-Nearest Neighbors: A moderate improvement is visible up to order 2 (AIC = 5136.5), but the indicator then increases (5244.7 and then 5314.1), revealing potential overfitting from order 3 onward.One-Class SVM: By contrast, the AIC continues to rise markedly with the Markov order, from 5918.5 to 8524.8. This behavior highlights the inadequacy of this model for temporal dynamics, made worse by the induced complexity.Robust Covariance: No significant variation is observed, with a constant AIC around 8372 for all orders. This confirms that this model does not exploit sequential memory, being purely based on the static structure of the covariance matrix.

The introduction of Markovian memory markedly influences the overall performance of the models. Figure 5 (Evolution of F1-score) shows that order 2 constitutes an optimal zone, where the F1-score reaches its maximum for most of the algorithms tested. This improvement is accompanied by an increase in recall, meaning better anomaly detection (reduction in false negatives), while maintaining acceptable precision.

Structural memory models, such as MLP, IF, or Random Forest, clearly benefit from this short-term memory. Their performance stagnates or slightly declines beyond order 2, reflecting a saturation effect. Conversely, models sensitive to noise, such as OCSVM or Robust Covariance, see their performance decrease with increasing order. This is visible in the confusion matrices (Figure 3: Confusion Matrices), where false positives increase significantly at orders 3 and 4, indicating degraded generalization.

## 5. Explainability and Causality

In this section, we analyze the results in terms of explainability and causal relationships. The objective is to highlight, through SHAP values and causal graphs, the specific contribution of each feature as well as the key interactions that influence the decision-making process of the hybrid models. The visualizations presented thus allow for a better understanding of the underlying logic of detections, as well as the impact of sequential memory integration on the structure of dependencies between variables.

### 5.1. Evolution of Explainability

The study of local importances, combined with a global view of memory impact across all models through SHAP, provides insight into the effect of memory on interpretability. Figure 9 reveals that order 2 introduces significant new variables related to previous states. At the second order, temporal statistics appear at the top of the list, indicating that the model usefully exploits the previous context to characterize anomalies. Conversely, at order 4, the total importance is distributed more uniformly among many variables, a sign of unnecessary complexification: no new variable clearly dominates, making the explanation less clear. To better understand the contributions of different features on the decision-making of different models, we analyzed the SHAP values of features on detected anomalies for the MLP model. As illustrated in Figure 10, the variables “pktTotalCount” and “octetTotalCount” maintain a predominant role, displaying standard deviations of 2.25 and 1.73, respectively, which testifies to their ability to significantly influence the model output. Other features such as “log_byte_rate, log_packet_rate, flowDuration, packet_rate, byte_rate, and count” also present respective standard deviations of 0.70 to 1.43, highlighting the richness of signals exploited even in the absence of sequential memory.

The proportion of positive contributions % SHAP > 0 oscillates between 51% and 61% depending on the variable (Table 9), which translates to a moderate directional effect: the variables do not systematically drive the model’s decision in one direction but participate in a balanced manner in the activation of anomaly prediction. The means of SHAP values, all close to zero, further confirm this balance. The stateless model manages to leverage traffic statistical indicators to explain its decisions, but the dispersion and intensity of SHAP contributions remain moderate. This suggests that, although effective, the purely statistical approach does not capture all the sequential complexity inherent to the industrial context.

Furthermore, the integration of the sequential memory of order 1 (Figure 11) marks an intermediate step in the explicative progression of the model.

There is a moderate increase in the variance of SHAP contributions for most variables, particularly for « octetTotalCount » σ = 4.40 and « pktTotalCount » σ = 4.00, while the variable “markov_score_order1” reaches a standard deviation of 3.00 and is already among the main sources of explanation (Table 10). This demonstrates the model’s ability to exploit local dependencies, relying on short-term traffic dynamics to refine anomaly prediction.

The proportions of positive contributions remain high for the structuring variables, with « octetTotalCount » at 65.0% and « flowDuration » at 64.8%, reflecting a clear directional effect: high values continue to favor anomaly detection. At the same time, the increased dispersion of SHAP values, particularly for the Markov order 1 score, indicates greater diversity in the situations interpreted by the model. In concrete terms, this means that the model mainly detects anomalies based on: (1) an abnormally high volume of data exchanged; (2) an unusual sending rate; and (3) an unexpected sequence of messages.

On the other hand, the appearance of the Markov order 2 score (Figure 12) in fourth place among the important features illustrates the model’s ability to exploit sequential and contextual patterns, which are often undetectable by conventional statistical metrics alone.

This use of sequential memory size at order 2 brings about a change in the distribution and the importance of SHAP values. First, there is a notable increase in the variance of contributions for all variables, particularly « octetTotalCount » σ = 4.74, « pktTotalCount » σ = 4.24, and now “markov_score_order2” σ = 3.31 (Table 11). This increase in the importance of the Markov score illustrates the model’s ability to exploit sequential and contextual patterns, which are often undetectable by statistical metrics alone. The proportion of positive contributions increases for most variables, notably « octetTotalCount » 64.2% and « flowDuration » 66.8%, reflecting a reinforced directional effect: high values along these dimensions more frequently pull the predictions toward the anomaly. In addition, the greater dispersion of SHAP values for the main sequential counters and scores suggests that the influence of each observation is modulated by its temporal or sequential context, enabling the model to more finely discriminate between risky situations.

In summary, the comparison of the three configurations clearly highlights the progressive contribution of sequential memory to the model’s explainability and performance. While the stateless model effectively exploits statistical variables, the successive introduction of first-order and then second-order Markov scores enhance the model’s ability to capture system dynamics, enriches the diversity of explanatory contributions, and improves the accuracy of anomaly detection. This increase in power confirms the value of a hybrid approach combining classical statistics and sequential memory for the analysis of industrial systems.

### 5.2. Evolution of the Causal Structure According to Memory Order

The causal structure graphs obtained for the MLP model explicitly illustrate the evolution of the complexity of the dependencies within the data as the Markov memory order integrated into the detection pipeline is increased. Without sequential memory (see Figure 13), anomaly detection relies essentially on direct relationships between the statistical variables and the anomaly target variable, which limits the depth of analysis of the model.

The addition of a second-order Markov memory (Figure 14) significantly complicates the causal structure: new intermediate nodes (Markov_1_, Markov_2_) appear, and the number of conditional dependencies linking statistical and sequential variables increases.

These intermediate states act as mediators, transmitting the joint influence of variables such as « service » and « log_packet_rate » to the final anomaly decision. Such a structure demonstrates the model’s ability to capture temporal dependence and changes in network flows, which is essential for detecting multi-step attack scenarios. Beyond order 4. (Figure 15), the graph becomes saturated with parasitic interactions: a multitude of weak links emerge between old and new variables without obvious causal justification, a sign of excessive variability. This phenomenon corroborates the idea that a high order “masks” the real links. This proliferation of edges suggests that an order higher than 2 “conceals” causal relationships, making inference more uncertain amid amplified spatio-temporal noise. The comparison with the causality graph confirms that Markov–ML hybridization reproduces much of the multivariate relational richness while preserving interpretability. The causal analysis further highlights the central role of rate variables and their logarithmic transformations in the propagation of information toward the anomaly state.

## 6. Deployment in Edge-IIoT Environments

We evaluated the Edge-IIoT implementation in the proposed approach by measuring the chain: ingestion → feature calculation → inference → verdict. We developed a physical testbed to emulate an industrial Edge-IoT scenario (Figure 16). The setup integrates a Raspberry Pi 4 with a quad-core Cortex-A72 1.5 GHz CPU and 4 GB RAM, used to deploy the trained models on a typical edge computing device. In addition, a laptop is employed for simulation and telemetry injection while running the OPC UA client, and a PLCnext controller acting as an OPC UA server connected through a network switch. This architecture: a Raspberry Pi 4 with a quad-core Cortex-A72 1.5 GHz CPU and 4 GB RAM (Figure 15), is representative of edge gateways deployed in current industrial cyber-physical systems [87,88]. The objective is to evaluate whether our approach can work in real time locally without exceeding the computing and memory capacities of this embedded device.

The tests focus on the two configurations of our pipeline: without Markovian memory and with a second-order Markov chain, to measure the potential impact of adding temporal dependencies on embedded performance. On the one hand, since the Markovian score calculation is performed locally at each new transition observed, it avoids a complete reconstruction of the sequence and adds only a negligible additional cost in terms of computation ~2% of total FLOPs. The reference models were implemented with scikit-learn [89]. ONNX Runtime was used to optimize native calculations using NEON vectorizations [90], and the TREELITE library was used for the efficient execution of compact tree forests [91].

The test data consisted of simulated telemetry streams at rates ranging from 100 to 1000 messages per second, reproducing the load of a typical industrial network. Performance was evaluated according to several metrics. We determined the theoretical number of FLOPs per inference and the p95 latency in milliseconds on a batch of 10^5^ inferences. To measure latency, we instrumented the execution by recording high-resolution timestamps via time.perf_counter_ns in Python (3.13.6) and internal hooks in ONNX mode for each inference, then calculated the percentile over all 10^5^ measurements. The choice of p95 as an indicator is justified by its common use in Edge/Cloud SLAs [92], ensuring that 95% of requests run below this critical threshold rather than relying on an average that could obscure rare latency spikes. Our results were then compared to a strict Edge-IIoT SLA: latency < 10 ms, RAM < 150 MB, and CPU usage < 1 core. Although the Raspberry Pi 4 can theoretically achieve between 12 and 48 GFLOPs/s in single precision (NEON + FMA) [93], actual execution is limited by the degree of usable vectorization and memory access latencies. There is therefore a gap between the computational load and the measured latency, especially for models where calculations cannot be fully vectorized [94,95]. From a memory perspective, the MLP, Isolation Forest, and Random Forest models remain compact with a RAM footprint of <120 MB in Python, reduced to <25 MB in optimized ONNX format. Table 12 summarizes the overall results and details the performance obtained for each model considered. The MLP, Random Forest, and Isolation Forest models were favored because they combine fast inference, statistical robustness, and reduced memory footprint, which are essential criteria in the Edge-IIoT context [96,97,98]. On the other hand, k-NN suffers from linear complexity in memory and computation, and One-Class SVM (RBF) requires many support vectors, making their execution prohibitive on embedded platforms [99].

The MLP exhibits a p95 latency of approximately 1 ms in Python, reduced to below 0.5 ms after ONNX conversion. The addition of sequential memory preserves the F1-score almost entirely, with a variation limited to ΔF1 = 0.01. These results indicate that a small neural network can effectively exploit sequential context at negligible cost, making it a particularly promising candidate for edge deployment. The Random Forest is characterized by very low FLOPs complexity—on the order of only a few thousand comparisons—and a stable p95 latency around 1.5–2 ms in Python, reduced to 1.0–1.5 ms with ONNX. The order of Markov memory has no significant effect on either latency or accuracy, since forests are primarily constrained by memory access and branching operations rather than floating-point computations. The Isolation Forest exhibits behavior very similar to that of the Random Forest in terms of both latency and computational load, since it is likewise an ensemble of trees that evaluate a limited number of conditions. The incorporation of sequential features produces no significant improvement; however, it also introduces no degradation, thereby confirming the lightweight and stable nature of this model in Edge-IIoT scenarios. A direct comparison between Random Forest and Isolation Forest highlights two key points. First, both models are non-FLOPs-bound: their computational load is dominated not by multiplications and additions but by logical comparisons, memory accesses to tree nodes, and pointer chasing during recursive traversal [100]. Consequently, their operational cost is more accurately expressed in terms of the number of comparisons—approximately 2000–3000 for Random Forest with 200 trees, and around 1000 for Isolation Forest—than in FLOPs.

## 7. Discussions

We now discuss the key lessons learned from these experiments. Our results confirm that integrating sequential memory via a Markov chain enhances the robustness of anomaly detection in OPC UA networks. By modeling the normal order of operations, the system identifies attacks that would escape purely statistical models. The experiments also show that an order 1 or 2 is sufficient to detect attacks, with a higher K providing only a marginal gain and even increasing false positives during rare legitimate variations (Figure 5). The Markov model implicitly assumes the independence of flows; however, an attacker could spread their abnormal behavior over several sessions to circumvent detection. Considering inter-flow correlations or introducing aggregate metrics at the global level is therefore necessary to counter this type of evasion.

Interpretability and cause and effect: Joint analysis of the results of explainability and causal inference shows that adding a sequential memory of order 1 or 2 is a real asset for model decision-making, particularly MLP. This combination of SHAP and objective causality, as well as the addition of short memory, enriches the model without complicating it: the inclusion of order 1 or 2 memory improves anomaly detection by capturing sequence breaks that are invisible to a strictly static model. Conversely, the introduction of long memory (order 3 or 4) tends to introduce noise and dilute the relevance of the causal relationships identified, confirming that there is an optimal compromise between sequential richness and model robustness. The test on Raspberry Pi 4 demonstrated that learned parametric methods such as MLP or tree forests offer an ideal compromise for edge, thanks to their predictable computational load and fixed memory footprint, while being able to take advantage of additional sequential context variables. On the other hand, non-parametric “example-based” methods such as OCSVM and k-NN, although effective in terms of detection in other contexts, encounter scaling issues on constrained hardware, mainly due to exorbitant memory consumption that disqualifies them for scalable datasets. From an industrial cybersecurity perspective, these results are encouraging. They indicate that it is possible to perform sophisticated analyses in real time, directly in the field, to identify abnormal behavior, without exceeding the constraints of a small, embedded gateway. The latency of less than 5 ms achieved for the best algorithms means that an IIoT monitoring system can detect and potentially respond to incidents almost instantly, without necessarily offloading the computation to the cloud. In addition, the use of optimized runtimes (ONNX, Treelite) has proven to be crucial in achieving this performance: it has made it possible to fully exploit the hardware capabilities of the ARM Cortex-A72 processor, reducing the memory occupied by the models by an order of magnitude compared to a naive Python execution and significantly reducing latency. This highlights the importance for practitioners of porting detection models to efficient executable formats before deploying them on site [101]. Furthermore, the low memory footprint after optimization suggests that deployment on even more limited devices is feasible. For example, optimized versions (weight quantization, use of fixed-point inference) could be implemented on microcontrollers or industrial nano-computers. In our case, if we were targeting an ultra-constrained microcontroller-type platform (e.g., ESP32), one possible strategy would be to embed only the Markovian component of our approach (which requires only a few kilobytes of memory and very simple calculations), as discussed in the Section 7. In any case, on a standard edge gateway such as the Raspberry Pi, the complete integration of our pipeline proved to be functional and efficient, validating its ability to evolve in real-time industrial environments.

To situate our contribution in relation to the state of the art, we offer a summary comparison with several reference approaches for anomaly detection in industrial systems (Table 13). Unlike purely statistical methods such as Isolation Forest, Robust Covariance, or deep sequential models such as LSTM and GRU, which offer great expressiveness but at the cost of high memory footprint and risk of overfitting, our hybrid Markov–ML pipeline combines the capture of short sequential regularities with the robustness of detectors. This combination preserves low latency and demonstrates generalizability on OPC UA streams, while keeping computational complexity low for edge deployment. The novelty therefore lies in the controlled introduction of Markov memory into existing stateless models, offering a balanced compromise between temporal fidelity, computational efficiency, and applicability in edge environments.

Although the Markov–ML hybrid framework demonstrates a tangible improvement in anomaly detection in industrial environments, several limitations are worth noting. First, the experiments are based on public datasets, whose representativeness with respect to heterogeneous industrial deployments is debatable; proprietary data or data from real test benches could reveal behaviors not captured here. Second, the integration of Markov memory increases computational complexity, which can become a limiting factor in resource-constrained edge environments, despite the encouraging results obtained on Raspberry Pi. Third, the calibration of hyperparameters and decision thresholds, although supported by rigorous cross-validation, remains sensitive to data distributions and may require adjustments when transferred to other industrial contexts. Finally, the models tested do not yet incorporate online adaptation or conceptual drift detection mechanisms.

## 8. Conclusions and Perspectives

This paper presents a comprehensive approach to anomaly detection in OPC UA industrial networks, combining sequential modeling using adaptive Markov chains and classical machine learning models, all enhanced by explainability and causal inference modules. Experiments conducted on a representative simulated dataset (DoS attacks, MITM, impersonation) have demonstrated that the integration of sequential memory significantly improves detection performance, particularly for identifying attacks that exploit message order or timing. We have also shown that the hybrid approach maintains a high level of interpretability: thanks to SHAP, each alert can be explained in terms of business indicators (volume, frequency, etc.), which is essential for deployment in a real industrial environment. In addition, causal graph analysis provides a macro view of the behavior of the system under attack, which is potentially useful for diagnosis and root-cause traceability. Beyond this analytical dimension, the proposed approach also opens concrete perspectives for deployment in real industrial IoT environments. By combining sequential memory with explainable ML models, the proposed pipeline addresses two major field constraints: low latency and robustness against heterogeneous and evolving data streams. Its integrability on edge devices such as Raspberry Pi demonstrates that advanced detection can be embedded directly at the process level without relying exclusively on the cloud, thereby reducing both the attack surface and remediation delays. More broadly, this contribution aligns with the trajectory of Industry 5.0 [102], where cybersecurity must be native, distributed, and interpretable. It thus constitutes a milestone toward resilient and autonomous IIoT architectures capable of withstanding increasingly sophisticated industrial threats.

This work opens several avenues for exploration. First, it would be relevant to examine the generalizability of the proposed approach to real industrial environments, including more complex architectures combining data from sensors, actuators, and heterogeneous network flows. Furthermore, extending this approach to other structured industrial communication protocols beyond OPC UA appears to be a promising avenue, provided that the sequential models and extracted descriptors are adapted to the semantics specific to each protocol. This methodological diversification would not only enhance the performance of detection models but also help to fill a significant gap in the literature, as the available datasets are still very limited in terms of multiple protocols. However, these protocols often have explicit sequential dimensions, which are highly relevant for contextual anomaly detection in industrial environments.

## Figures and Tables

**Figure 1 sensors-25-06122-f001:**
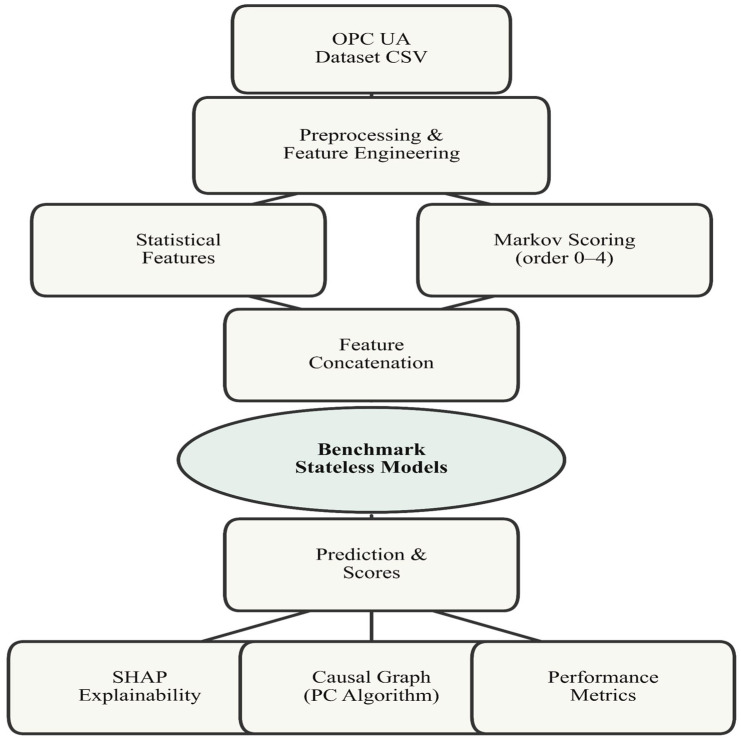
Overall architecture.

**Figure 2 sensors-25-06122-f002:**
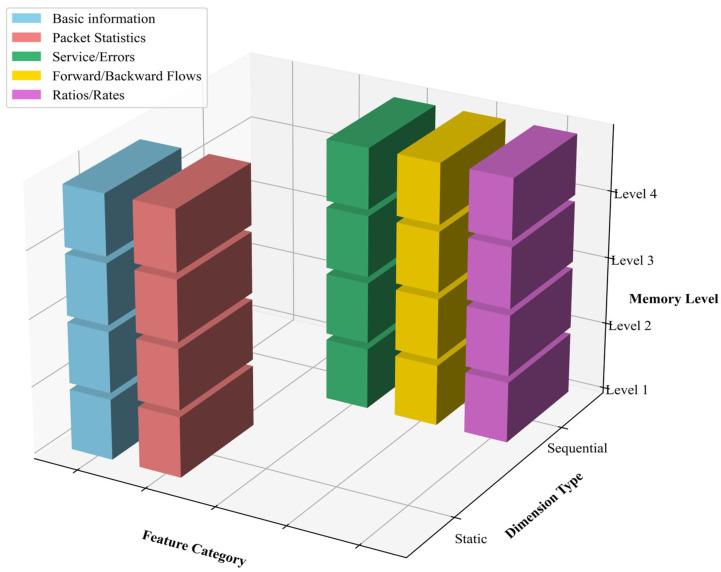
3D Matrix of OPC UA variables (Static vs. Sequential × Memory Levels).

**Figure 3 sensors-25-06122-f003:**
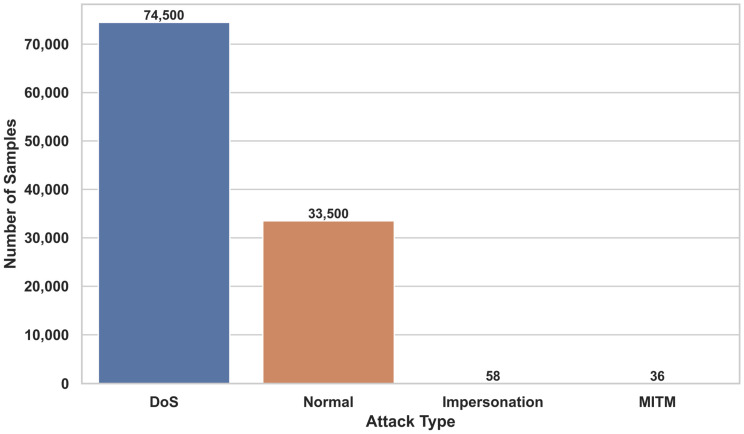
Distribution of labels in the OPC UA dataset.

**Figure 4 sensors-25-06122-f004:**
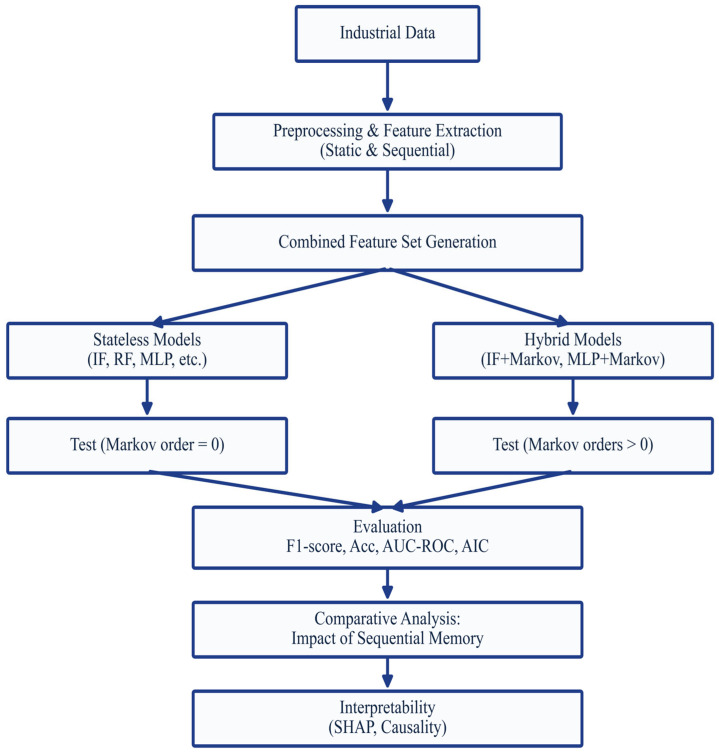
Experimental hybrid pipeline (stateless vs. Markovian).

**Figure 5 sensors-25-06122-f005:**
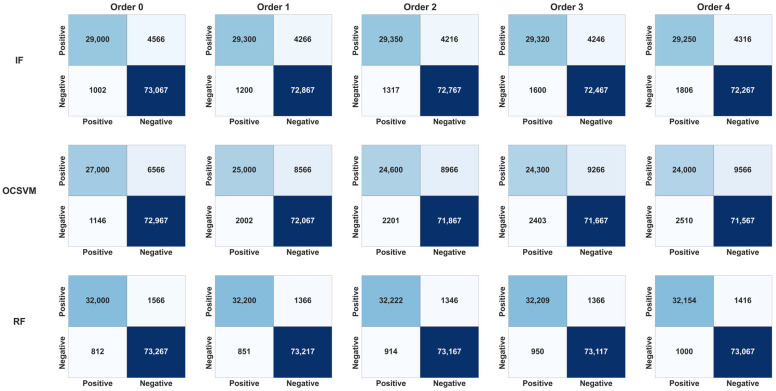
Confusion matrices of ML models (IF, OCSVM, and RF) according to the Markov memory order.

**Figure 6 sensors-25-06122-f006:**
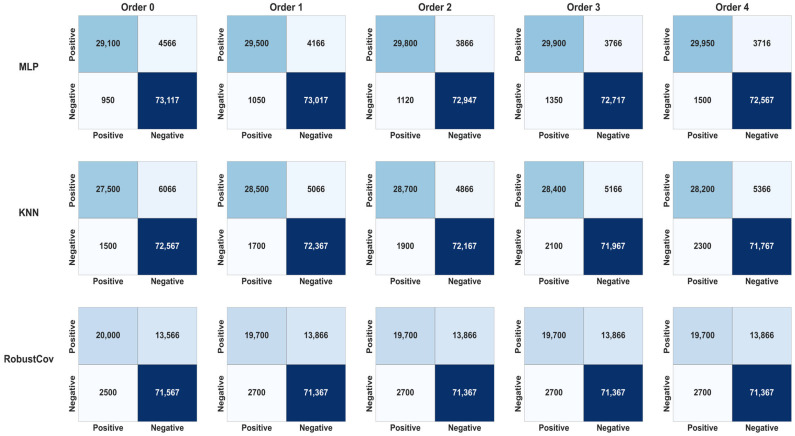
Confusion matrices of ML models (MLP, KNN, and RobustCov) according to the Markov memory order.

**Figure 7 sensors-25-06122-f007:**
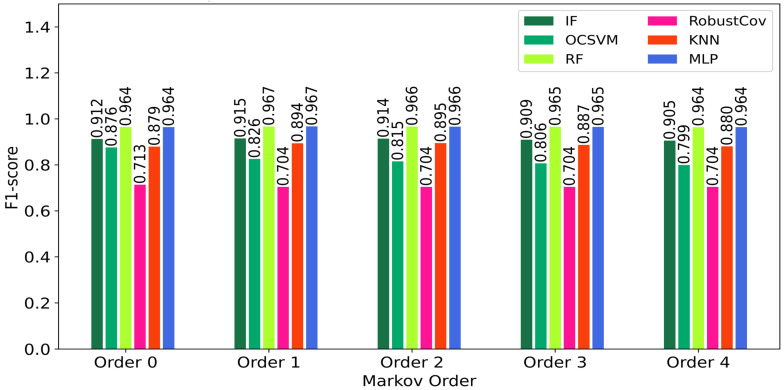
Impact of Markov order on the F1-score of ML Models.

**Figure 8 sensors-25-06122-f008:**
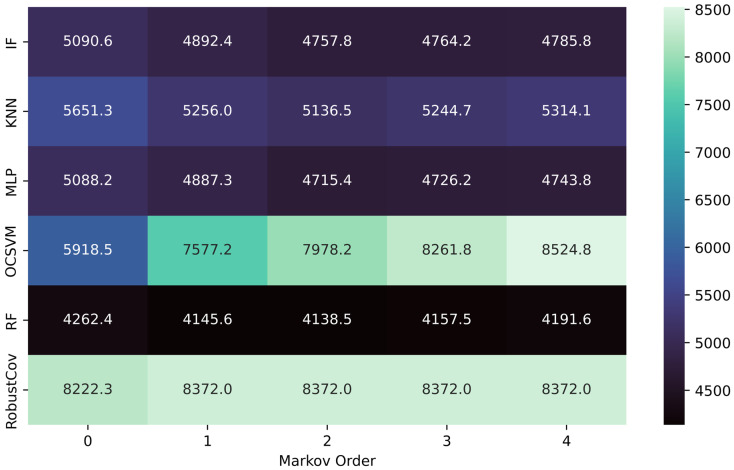
Heatmap of the Akaike Information Criterion (AIC).

**Figure 9 sensors-25-06122-f009:**
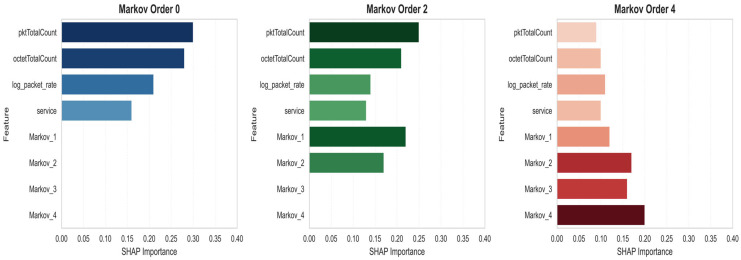
Distribution of SHAP importance by Markov order.

**Figure 10 sensors-25-06122-f010:**
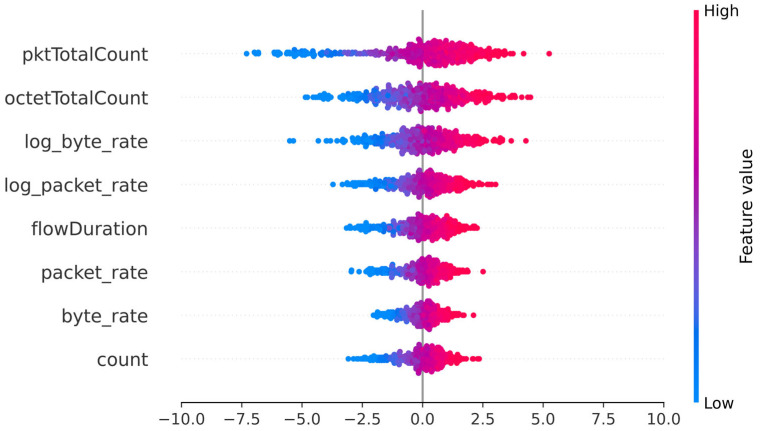
Analysis of SHAP values for the stateless MLP.

**Figure 11 sensors-25-06122-f011:**
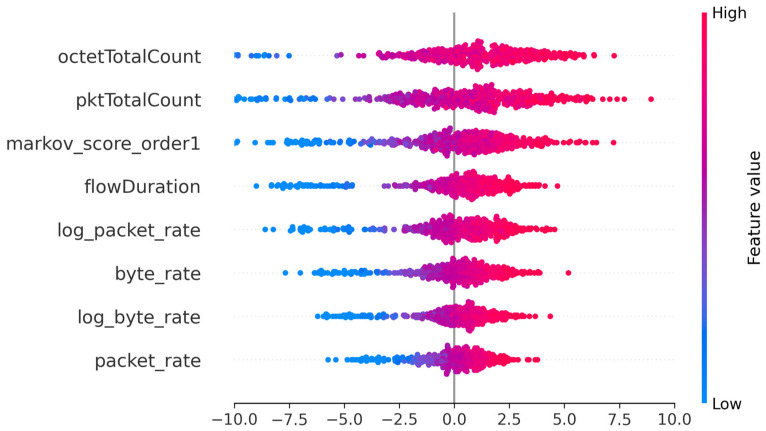
Analysis of SHAP values on the MLP model with Markov order 1.

**Figure 12 sensors-25-06122-f012:**
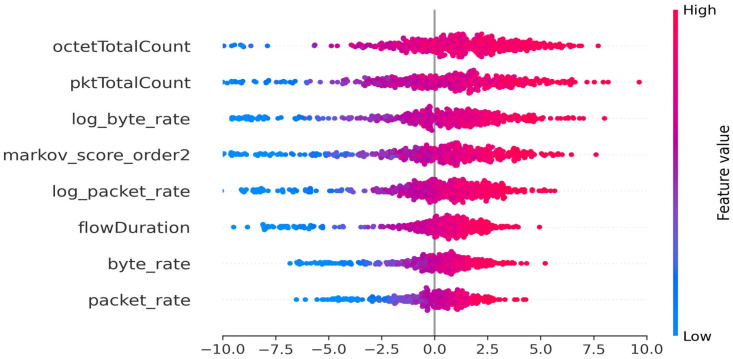
Analysis of SHAP values on the MLP model with Markov order 2.

**Figure 13 sensors-25-06122-f013:**
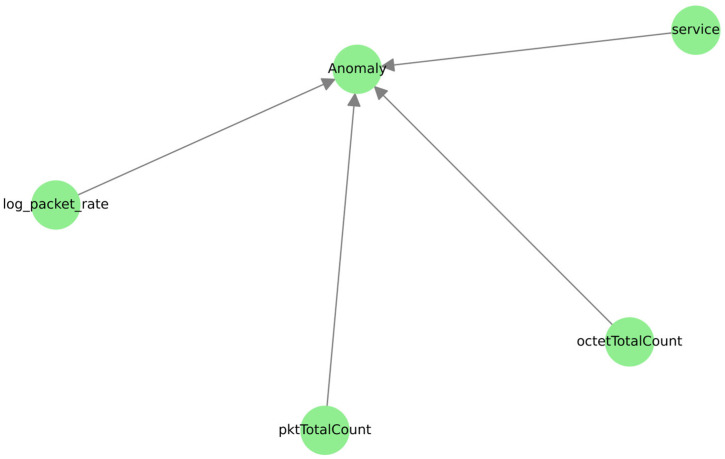
Graph of anomaly detected by stateless MLP.

**Figure 14 sensors-25-06122-f014:**
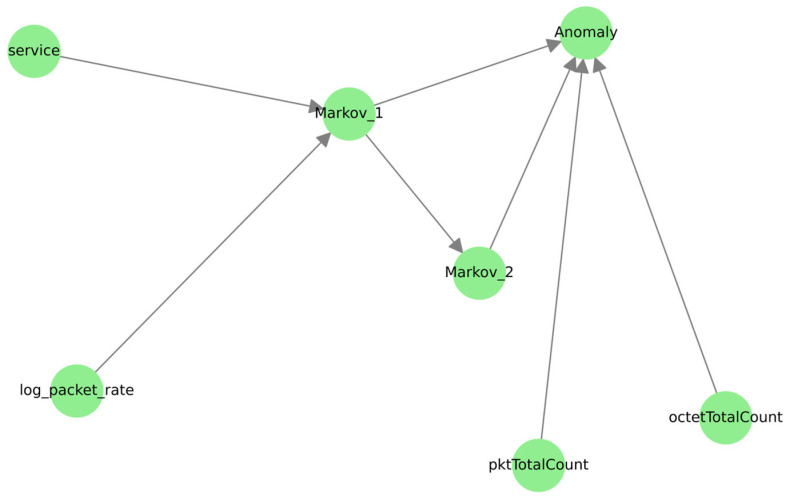
Graph of anomaly detected by MLP with Markov order 2.

**Figure 15 sensors-25-06122-f015:**
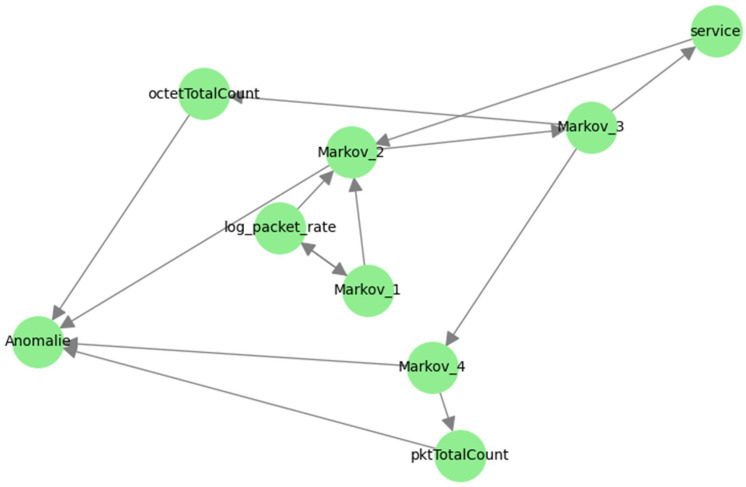
Graph of anomaly detected by MLP with Markov order 4.

**Figure 16 sensors-25-06122-f016:**
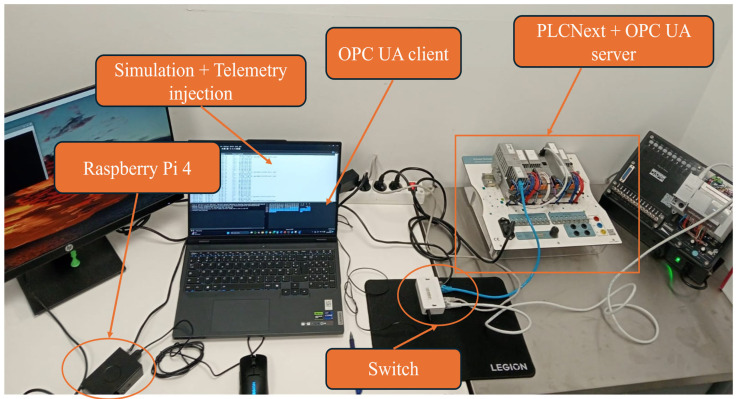
Test environment for simulated telemetry injection and OPC UA data exchange.

**Table 1 sensors-25-06122-t001:** Thematic distribution of network variables in the OPC UA dataset.

Throughput/Load	Timing/Duration	Business Ratios	Errors/Message Sizes
- pktTotalCount - octetTotalCount - avg_ps - f_rate - b_pktTotalCount - f_pktTotalCount	- flowDuration - flowInterval - avg_flowDuration - f_flowStart - b_flowStart	- same_srv_rate - dst_host_same_src_port_rate - count, srv_count	- msg_size - min_msg_size - service_errors - status_errors

**Table 2 sensors-25-06122-t002:** Evolution of Random Forest performance.

Markov Order	F1-Score	Precision	Rappel	Difference F1 (%) vs. Order 0
0	0.964	0.976	0.953	0%
1	0.967	0.974	0.959	+0.4
2	0.966	0.973	0.960	+0.2%
3	0.965	0.971	0.959	+0.1%
4	0.964	0.970	0.958	0%

**Table 3 sensors-25-06122-t003:** Evolution of Isolation Forest performance.

Markov Order	F1-Score	Precision	Rappel	Difference F1 (%) vs. Order 0
0	0.912	0.967	0.864	0%
1	0.915	0.961	0.873	+0.25%
2	0.914	0.958	0.874	+0.18%
3	0.909	0.948	0.874	−0.36%
4	0.905	0.942	0.871	−0.8%

**Table 4 sensors-25-06122-t004:** Evolution of K-Nearest Neighbors performance.

Markov Order	F1-Score	Precision	Rappel	Difference F1 (%) vs. Order 0
0	0.879	0.948	0.819	0%
1	0.894	0.944	0.849	+1.7%
2	0.895	0.938	0.855	+1.8%
3	0.887	0.931	0.846	+0.9%
4	0.880	0.928	0.834	+0.1%

**Table 5 sensors-25-06122-t005:** Evolution of Multi-Layer Perceptron performance.

Markov Order	F1-Score	Precision	Rappel	Difference F1 (%) vs. Order 0
0	0.964	0.9756	0.953	0%
1	0.967	0.9743	0.959	+0.31%
2	0.966	0.9728	0.960	+0.21%
3	0.965	0.9713	0.959	+0.10%
4	0.964	0.9698	0.958	0%

**Table 6 sensors-25-06122-t006:** Evolution of One-Class SVM model performance.

Order Markov	F1-Score	Precision	Rappel	Difference F1 (%) vs. Order 0
0	0.876	0.961	0.804	0%
1	0.826	0.926	0.745	−5.7%
2	0.815	0.918	0.733	−7.0%
3	0.806	0.910	0.724	−8.0%
4	0.799	0.906	0.715	−8.8%

**Table 7 sensors-25-06122-t007:** Evolution of Robust Covariance model performance.

Markov Order	F1-Score	Precision	Rappel	Difference F1 (%) vs. Order 0
0	0.713	0.889	0.596	0%
1	0.708	0.879	0.587	−1.3%
2	0.704	0.879	0.585	−1.6%
3	0.702	0.874	0.582	−1.6%
4	0.704	0.872	0.580	−1.8%

**Table 8 sensors-25-06122-t008:** Comparative performance of MLP, LSTM, and GRU.

Model	F1-Score	Precision	Rappel	RAM (MB)
LSTM	0.972	0.965	0.980	883
GRU	0.970	0.963	0.978	864
MLP + Markov-2	0.966	0.972	0.960	≈312

**Table 9 sensors-25-06122-t009:** Summary of the statistics of SHAP values for stateless models.

Features	% SHAP > 0	Mean	Standard Deviation
pktTotalCount	61.0%	−0.02	2.25
octetTotalCount	58.0%	0.01	0.90
log_byte_rate	57.2%	−0.01	0.92
log_packet_rate	54.8%	0.007	0.70
flowDuration	54.6%	0.01	1.02
packet_rate	54.2%	0.018	1.22
byte_rate	51.22%	−0.01	1.73
count	51.2%	−0.012	1.43

**Table 10 sensors-25-06122-t010:** Summary of the statistics of SHAP values for Markov order 1.

Features	% SHAP > 0	Mean	Standard Deviation
octetTotalCount	65.0%	−0.05	4.40
pktTotalCount	64.8%	0.01	2.54
markov_score_order1	60.2%	−0.01	1.81
flowDuration	58.8%	0.02	2.01
log_packet_rate	58.6%	0.02	2.36
byte_rate	58.4%	−0.04	4.00
log_byte_rate	58.0%	−0.01	3.00
packet_rate	57.6%	−0.02	1.53

**Table 11 sensors-25-06122-t011:** Summary statistics of SHAP values for Markov order 2.

Features	% SHAP > 0	Mean	Standard Deviation
octetTotalCount	66.8%	0.03	2.51
pktTotalCount	64.2%	−0.06	4.74
log_byte_rate	59.2%	−0.01	3.58
markov_score_order2	59.0%	0.02	3.31
log_packet_rate	58.8%	0.03	2.89
flowDuration	58.6%	−0.04	4.24
byte_rate	58.4%	−0.01	2.01
packet_rate	57.6%	−0.02	1.74

**Table 12 sensors-25-06122-t012:** Inference performance on Raspberry Pi 4.

Model	Order	F1_Off	F1_Edge	ΔF1	p95 Py (ms)	p95 ONNX (ms)	Flops
MLP	0	0.964	0.959	−0.005	1.02	0.54	11.12 k
MLP	2	0.966	0.962	−0.004	1.49–1.98	0.93–1.51	11.43 k
RF	0	0.961	0.959	−0.002	1.51–2.03	1.04–1.50	3 k comparisons
RF	2	0.963	0.961	−0.002	3.61–3.92	2.5–2.90	3 k comparisons
IF	0	0.912	0.909	−0.003	4.03–5.08	3.09–3.48	1 k comparisons

**Table 13 sensors-25-06122-t013:** Comparative evaluation of models for Edge-IIoT deployment.

Ref	Approach	Sequentially	Latency/Resources	Edge Applicability
[14]	Autoencoder	++	–	–
[34,35]	LSTM/GRU	++	–	–
[64]	Isolation Forest	–	+	++
[65]	Random Forest	–	±	+
[73]	MLP	–	+	++
Our approach	Markov–ML	+(K=2)	+	++

Index notation: Sequentially: – = stateless, + = sequence-aware, and ++ = native sequential. Latency/Resources: – = very costly, ± = moderate latency, and + = lightweight. Edge Applicability: – = limited, + = optimizable, and ++ = excellent.

## Data Availability

The data presented in this study are available on request from the corresponding author.

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
