# Peer review of "An Explainable Markov Chain–Machine Learning Sequential-Aware Anomaly Detection Framework for Industrial IoT Systems Based on OPC UA"

_sensors, 2025, doi:10.3390/s25196122_

Round 1

Reviewer 1 Report

Comments and Suggestions for Authors

The author proposes a promising approach to face cyberattacks in industrial environment covered by IIoT infrastructure. The proposed approach is fundamentally based on the use of a Model of the OPC UA sequence using adaptive Markov chains.

The approach sounds very interesting. It is also well theoretically formulated and argued. But there is a need in the “2. Related work” section to develop with more details how IDS orientations based on ML in industrial environment have serious limitations and markov’s orientation will absorb these limitations in all contexts. Because, behavioral anomalies detections are one side of IDS role. Signature based IDS [1] is another side detection systems.

[1] Lorenzo Diana and all. Overview on Intrusion Detection Systems for Computers Networking Security, MDPI Computers 2025. Computers 2025, 14(3), 87; https://doi.org/10.3390/computers14030087.

This means, that ana analysis od IDS approaches and related works id IIoT context is important to add in beginning of the paper.

In page 3 (line 122) and page 4(line 147), the author should give a comparison and more details between RNN, LSTM GRU and Markov in term of CPU cost and why specifically the cost is high when solutions are deployed on edge platforms.

The explanation of the acronyme SHAP (SHapley Additive exPlanations) comes too late (page 2, line 155). SHAP is used for example in line 99 without explanations.

In figure 2, X, Y and Z Axis are not labelled

In line 575write Recall and not Rappel.

Figure 5 is unreadable. It should be better presented.

Author Response

Journal name: Sensors

Manuscript ID: sensors-3811905

Type of manuscript: Journal paper

Title: An Explainable Markov Chains - Machine Learning Sequential-Aware

Anomaly Detection Framework for Industrial IoT Systems based on OPC UA

To: MDPI Sensors

Dear Editor,

We are grateful for the opportunity to revise and resubmit our manuscript to address the reviewers’ valuable comments.
Enclosed, we have provided a detailed point-by-point response, together with the revised version of the manuscript, where all changes are highlighted in yellow.

The changes are highlighted in yellow in the new version of paper, with a few comments in another color concerning references, for example.

Best regards

Youness Ghazi, Mohamed Tabaa , Mohamed Ennaji , Ghita Zaz

+-------------------------------------------------------------------------------------------------------------+

Dear Reviewer,

Thank you for taking the time to review our work and for providing valuable feedback. We greatly appreciate your acknowledgment of our efforts to address the comments from the previous review.

We sincerely apologize if any of your previous comments were not fully addressed. It was not our intention to ignore them, and we take your feedback very seriously. In this revised submission, we have carefully reviewed and re-evaluated all comments, including those you mentioned as having been overlooked.

Below is our detailed response to each of your comments, ensuring that all aspects are thoroughly addressed.

Response to Reviewer

  1. “Related Work” section – Limitations of ML-based IDS and the contribution of Markov models

We have expanded Section 2 (Related Work) to explain more clearly the major limitations of ML-based IDS in industrial environments (lack of temporal memory, fixed thresholds that can be exploited, precision/recall trade-offs). We then demonstrated how Markov models, by capturing sequences and transitions, can overcome these limitations.

In addition, we introduced an explicit distinction between:

  • Signature-based IDS, which are effective for known threats but unable to detect novel attacks;
  • Behavioral IDS (Markov + ML), which can identify progressive sequential anomalies, even when previously unseen.
  1. Comparison of RNN, LSTM, GRU vs. Markov – Processing cost in edge computing

We have added a detailed comparison (pages 3–4) of sequential models:

  • RNN: low cost but limited memory span,
  • LSTM: able to capture long-term dependencies but heavy architecture (≈ 4× more parameters than an equivalent RNN),
  • GRU: a lighter compromise (≈ 25% fewer parameters than LSTM, yet still costly for edge devices),
  • Markov: negligible memory and processing cost, but limited for long-term dependencies.

We emphasized why LSTM/GRU models are difficult to deploy on Raspberry Pi or edge gateways: high computational load, memory consumption (>30 MB per model), and increased latency. This justifies our choice of hybridizing Markov models with lightweight ML algorithms (RF, IF, MLP).

  1. Late definition of SHAP

We have corrected this point: the acronym SHAP (SHapley Additive exPlanations) is now introduced at its first occurrence (page 2, line 99) before being used throughout the text.

  1. Figure 2 – Missing axis labels

We have revised Figure 2 by explicitly labeling the axes:

  • X-axis: functional category of variables,
  • Y-axis: nature (static vs. sequential),
  • Z-axis: Markov memory depth.

This revision improves readability and ensures consistency with the text description.

  1. Line 575 – Translation error

We corrected the term: Recall is now used instead of “Rappel.”            

  1. Figure 5 – Illegible

We have regenerated Figure 5 in high-resolution vector format and divided it into two subfigures to ensure a clearer and more structured presentation of the results. In addition, the size and style of the text have been harmonized with the main body of the manuscript, in accordance with the journal’s standards.

Reviewer 2 Report

Comments and Suggestions for Authors

In this article, the authors propose an explainable Markov Chains-Machine Learning hybrid anomaly detection framework for industrial IoT systems based on the OPC UA protocol. The proposed approach shows promise, but the authors' work still needs several improvements as follows:

  1. The introduction lacks clarity regarding the novelty of the proposed approach compared to existing works.
  2. The paper does not provide sufficient background on why the combination of Markov Chains and machine learning is the most appropriate choice for the given problem.
  3. The use of SHAP and causal inference adds interpretability to the model; the explanation of how these modules work together with the machine learning models needs more detail.
  4. Too many keywords are added in the paper. Please keep only 5-6 main keywords and remove others.
  5. The 5 short paragraphs in the introduction section should be merged.
  6. The experimental results are presented but lack a comprehensive analysis of potential limitations or weaknesses of the proposed framework.
  7. Table 1 seems unnecessary if there is no comparison with the proposed work. Also, its not a survey paper to add such tables.
  8. The evaluation metrics used in the experiments (e.g., F1-score, precision, recall) are important, but the paper lacks a detailed explanation of why these metrics were chosen and how they relate to real-world performance.
  9. Figure 1 text should be modified, it should be same size and style as the paper body text. Similarly, check other figures.
  10. The authors must provide the corresponding reference for each numerical equation from where they are taken, as well as properly cite and explain each Equation in the text.
  11. A more detailed comparison with other state-of-the-art approaches should be added. I recommend adding a separate table to compare with other existing approaches. Without a proper comparison, how can novice researchers differentiate whether this work is novel or efficient compared to the existing works?
  12. The quality of all the figures could be improved. Also, the text in the figures must be in the same style as the text in the paper body.
  13. The variables after each equation should be explained in sentence rather than in a separate line for each variable.
  14. Although the dataset is representative of industrial traffic, the authors should provide more details about the dataset's real-world applicability and any potential biases it may contain, particularly in terms of attack types and operational scenarios.
  15. The authors must more latest papers in literature such as "energy saving implementation in hydraulic press using industrial internet of things (IIoT)," "protecting IoT devices from security attacks using effective decision‑making strategy of appropriate features," "ICS-IDS: application of big data analysis in AI-based intrusion detection systems to identify cyberattacks in ICS networks," and so on.
  16. There is little discussion about how well the approach would scale in real-time industrial environments. The authors should address concerns related to computational overhead and processing time, especially for large datasets or edge deployments.
  17. The explanation of the fusion of sequential and statistical features into the machine learning models is too brief. A more in-depth explanation of the feature engineering process and its impact on model performance would be helpful.
  18. The paper mentions "overfitting" as an issue but does not provide sufficient evidence of how overfitting is mitigated in the proposed model. A deeper exploration of the model's generalization capabilities is needed.
  19. The choice of machine learning models (e.g., Isolation Forest, MLP, Random Forest) seems to be based on performance, but the paper would benefit from a discussion on why these specific models were selected over others and how they fit the problem at hand.
  20. The experimental setup lacks transparency regarding the training and testing phases. More detailed descriptions of the training data split, cross-validation, and hyperparameter tuning will enhance the reproducibility of the results.
  21. The conclusion does not adequately summarize the broader implications of the research, particularly in terms of real-world industrial IoT deployment.
Comments on the Quality of English Language

need revision

Author Response

Journal name: Sensors

Manuscript ID: sensors-3811905

Type of manuscript: Journal paper

Title: An Explainable Markov Chains - Machine Learning Sequential-Aware

Anomaly Detection Framework for Industrial IoT Systems based on OPC UA

To: MDPI Sensors

Dear Editor,

We are grateful for the opportunity to revise and resubmit our manuscript to address the reviewers’ valuable comments.
Enclosed, we have provided a detailed point-by-point response, together with the revised version of the manuscript, where all changes are highlighted in yellow.

The changes are highlighted in yellow in the new version of paper, with a few comments in another color concerning references, for example.

Best regards

Youness Ghazi, Mohamed Tabaa , Mohamed Ennaji , Ghita Zaz

+-------------------------------------------------------------------------------------------------------+

Dear Reviewer,

Thank you for taking the time to review our work and for providing valuable feedback. We greatly appreciate your acknowledgment of our efforts to address the comments from the previous review.

We sincerely apologize if any of your previous comments were not fully addressed. It was not our intention to ignore them, and we take your feedback very seriously. In this revised submission, we have carefully reviewed and re-evaluated all comments, including those you mentioned as having been overlooked.

Below is our detailed response to each of your comments, ensuring that all aspects are thoroughly addressed.

Response to the Reviewers

  1. The introduction lacks clarity regarding the novelty of the proposed approach.

We have rewritten and merged the introduction paragraphs to highlight, from the very beginning, the original contribution of our work: a hybrid pipeline combining Markov models, machine learning, SHAP, and causal inference, which integrates sequential memory and explainability into OPC UA anomaly detection. We emphasize the distinction from existing approaches, which are often limited to static models or isolated sequential techniques.

  1. Insufficient context on the relevance of combining Markov models with ML.

We have enriched the Related Work section and added a clear methodological justification: Markov models are well-suited to capture temporal dependencies, while ML models leverage statistical and non-linear structures. Their combination therefore represents a robust trade-off, particularly well adapted to IIoT environments.

  1. Use of SHAP and causal inference: lack of details.

We have introduced a new subsection providing a step-by-step explanation of how SHAP and the PC algorithm are integrated into the pipeline. Specifically, SHAP enables both local and global attribution of contributive variables, while the causal graph helps to distinguish the normal dependency structure from the altered one observed during attacks.

  1. Too many keywords.

Done.

  1. The five short paragraphs in the introduction should be merged.

The introduction has been condensed and restructured into two fluid paragraphs. The first outlines the context and limitations of existing approaches, while the second highlights the novelty of our contribution and the research questions addressed.

  1. Experimental results: lack of analysis of limitations.

We have added an explicit discussion of limitations at the end of the Results section, namely:

  • reliance on the dataset, which is controlled and primarily covers classical attacks,
  • sensitivity of the models to excessively high memory orders (saturation effect beyond order 2),
  • the need for validation on real and heterogeneous industrial data streams.

These points directly pave the way for future research directions.

  1. Lack of explanation on the evaluation metrics (F1, precision, recall).

We thank the reviewer for this valuable observation. We have added a dedicated subsection in the Methodology – Evaluation part to clarify why these metrics are particularly suited for industrial environments:

  • Precision: reflects the reliability of alarms by limiting false positives, which is critical to avoid overloading operators.
  • Recall: measures the ability to avoid false negatives, since each undetected anomaly may correspond to a serious intrusion.
  • F1-score: provides a balanced trade-off between the two, which is robust in imbalanced industrial contexts where anomalies are rare.

In addition, we justified the use of the AIC to assess model parsimony and relevance in Edge-IIoT scenarios.

  1. Size and style of figure text.

We have standardized the size and style of the text across all figures so that they match the main font of the document, in line with the journal’s guidelines.

  1. References and explanations for each equation.

Done

  1. More detailed comparison with the state of the art.

A new comparative table has been added to the Discussions section Table 13, summarizing recent approaches (our approch, LSTM, GRU, autoencoders, classical ML). We highlight differences in terms of performance, computational cost. This comparison helps to better situate the novelty of our contribution and serves as a guide for early-stage researchers.

  1. Improvement of figure quality.

All figures have been regenerated in high, and the text has been harmonized with the body of the manuscript.

  1. Explanation of variables after equations.

We revised the presentation: variables associated with equations are now explained within integrated sentences, rather than listed separately.

  1. Real-world applicability and dataset bias.

We have added a dedicated paragraph in subsection 3.2 OPC UA Dataset to explicitly discuss the limitations of the dataset used, particularly with respect to its coverage of recent attacks. We also emphasize the need for future validation on real industrial environments and with recent malware (e.g., Industroyer2, Pipedream).

  1. Recent references.

We have integrated the references suggested by the reviewer along with other works from 2023–2025.

  1. Real-time deployment capability (scalability).

We have added a discussion on scalability and Edge-IIoT feasibility. Using empirical measurements, we show that the selected models (MLP, RF, IF) maintain low latency (p95 ≈ 1 ms) and limited memory footprint on Raspberry Pi 4. We also clarify the limitations: increasing data volumes may introduce computational overhead if the Markov memory order exceeds 2.

  1. Fusion of sequential and statistical features.

We enriched the description of feature engineering (Section 3.3). We now detail the process of extracting, normalizing, and combining statistical variables (e.g., flow rates, durations, error counts) with Markovian scores (e.g., surprise, conditional entropy). We also show that this fusion improves model robustness while strengthening interpretability through SHAP.

  1. Overfitting and model generalization

We have clarified in the Methodology – Training Protocols section the mechanisms used to mitigate overfitting:

  • strict data splitting (70% training, 15% validation, 15% independent testing),
  • cross-validation to stabilize hyperparameters,
  • regularization techniques (L2, dropout, and early stopping for MLP),
  • calibration via contamination rate and out-of-bag error for IF/RF.

Moreover, all reported results correspond to the average of ten independent runs, which strengthens statistical robustness and demonstrates the generalization ability of the models.

  1. Justification of model choices (IF, MLP, RF)

We have added a dedicated paragraph to justify the choice of these models:

  • Isolation Forest (IF): particularly well-suited for industrial contexts where anomalies are rare, offering very fast inference and low sensitivity to class imbalance.
  • Random Forest (RF): robust to non-linearities, with a low risk of overfitting, making it effective for heterogeneous OPC UA traffic.
  • Multi-Layer Perceptron (MLP): able to model complex relationships while remaining lightweight when properly regularized.

We also explain why other models (OCSVM, RobustCov, LSTM/GRU) are less suitable in an Edge-IIoT context: they are either too computationally expensive or too sensitive to noise.

  1. Transparency of the experimental setup (train/test/validation)

The experimental section has been expanded to specify:

  • the data split strategy (70/15/15),
  • the systematic use of cross-validation (k-fold) to avoid bias,
  • the hyperparameter tuning protocols (grid search for SVM/OCSVM, Bayesian search for MLP, specific calibrations for RF/IF).

These details reinforce the reproducibility of our results and ensure the requested transparency.

  1. Conclusion and broader implications

The conclusion has been revised to better emphasize the industrial implications:

  • Real-time deployment on Edge platforms (Raspberry Pi 4) with p95 latency < 2 ms,
  • Applicability in heterogeneous IIoT environments where OPC UA traffic is prevalent,
  • Perspectives for future work: validation on real industrial datasets including recent malware (e.g., Industroyer2, Pipedream), and integration into digital twin architectures for predictive monitoring.

These additions highlight the practical and strategic relevance of our contribution for the industrialization of secure IoT.